# Changes of cirrus cloud properties and occurrence over Europe during the COVID-19 caused air traffic reduction

Qiang Li[1] and Silke Groß[1]

[1]Deutsches Zentrum für Luft- und Raumfahrt, Institut für Physik der Atmosphäre, D-82234 Oberpfaffenhofen, Germany,

*Correspondence to:* Qiang Li (qiang.li@dlr.de) and Silke Groß (silke.gross@dlr.de)

**Abstract.** By inducing linear contrails and contrail cirrus, air traffic has a main impact on the ice cloud coverage and occurrence. During the COVID-19 pandemic the civil air traffic over Europe was significantly reduced: in March and April 2020 to about 80% compared to the year before. This unique situation allows to study the effect of air traffic on cirrus clouds. This work investigates based on satellite lidar measurements if and how cirrus cloud properties and occurrence changed over Europe in the course of COVID-19. Cirrus cloud properties are analyzed for different years between 2014 and 2019, which showed similar meteorological conditions for April as they were found for 2020. The meteorological conditions for March, however, were warmer and drier in 2020 than the previous years. The average thickness of cirrus clouds was reduced to 1.18 km in March 2020 compared to a value of 1.40 km under normal conditions, which is stronger than expected from the aviation reduction due to the less favorable meteorology for ice cloud formation. While the April results in 2020 were only slightly reduced with an average thickness 70 m thinner than the composite mean of the previous 6 years. Comparing the different years shows that the cirrus cloud occurrence was reduced by about 17–30% with smaller cloud thicknesses found in 2020 for both months. In addition, the cirrus clouds measured in 2020 possess smaller mean values of the particle linear depolarization ratio (PLDR) than the previous years at a high significance level for both months, especially at colder temperatures (T < -50 °C). The same exercises are extended to the observations over the United States of America and over China. Besides the regional discrimination of cirrus clouds, we reach the final summary that cirrus clouds show significant changes of PLDR in both March and April over Europe, no changes in both months over China, and significant changes only in April over USA.

## 1 Introduction

Cirrus clouds have a wide global coverage and thus a large effect on the Earth's radiation budget. It is assumed that mid-latitude cirrus clouds in general have a warming effect (*Chen et al.*, 2000), but their radiative effects strongly depend on their microphysical properties, e.g. particle number concentration, size, and shape (e.g. *Stephens et al.*, 1990; *Haag and Kärcher*, 2004). Ambient conditions, like temperature and supersaturation (e.g. *Heymsfield*, 1977; *Khvorostyanov and Sassen*, 1998), but also the nucleation mode (e.g. *Ström and Ohlsson*, 1998; *Seifert et al.*, 2004; *Urbanek et al.*, 2018) can influence the microphysical properties of the cirrus clouds. Previous studies reveal that ice crystals in air form and grow as a function of the ambient temperature and relative humidity and there is a general trend toward larger morphological complexity with increasing supersaturation at all temperatures (e.g. *Heymsfield*, 2003; *Bailey and Hallett*, 2004, 2009). Moreover, vertical wind velocities

are a key driver of ice nucleation in the atmosphere (e.g. *Shi and Liu*, 2016; *Kärcher and Jensen*, 2017; *Kärcher*, 2017). Based on laboratory experiments, Bailey and Hallett (2004) reported that different ice crystal habits were observed under conditions with different temperatures. The natural ice crystals, however, which encounter varying temperature and humidity may grow into irregular forms (*Korolev et al.*, 1999). Furthermore, mass transport (including convection and advection) and crystal origin at a sample region also govern the correlation between temperature and ice crystal habits (e.g. *Bailey and Hallett*, 2004; *Um et al.*, 2015). Differences in size and shape have an impact on the particles' optical properties; it was found that columnar ice crystals generate higher depolarization ratios than plate-like crystals (*Noel et al.*, 2006), with the highest lidar depolarization ratios found for irregularly shaped ice crystals.

According to theoretical ray-tracing simulations of laser backscatter depolarization (e.g. *Takano and Liou*, 1989), the geometric properties (shape and size) of aerosols and ice crystals have a strong influence on the scattering characteristics of light. Light scattering by atmospheric ice crystals lead to a change of polarization according to the internal ray paths, more precisely, increasing with increasing hexagonal axis ratio (= length over width). The particle linear depolarization ratio (PLDR) used to evaluate this effect is a well-defined parameter to retrieve information on ice crystal habits in terms of particle phase, shape, and orientation. The lidar transmits linearly polarized light into the atmosphere. The light scattered in backward direction by spherical particles has the same orientation of polarization as the incident light, whereas non-spherical particles such as cirrus ice crystals can display different polarization states according to their shape and size distribution (*Sassen et al.*, 1989; *Freudenthaler et al.*, 2009; *Urbanek et al.*, 2018). The polarization lidar technique is a well-established and widely-used method to provide information on aerosol profiling and to distinguish between different types of aerosols, e.g. non-spherical mineral dust particles with high values of the PLDR (*Freudenthaler et al.*, 2009; *Tesche et al.*, 2009; *Groß et al.*, 2012). It is also used to unambiguously differentiate between ice clouds and water clouds (e.g. *Bühl et al.*, 2016) and to study the characteristics of ice clouds (e.g. *Schotland et al.*, 1971; *Sassen*, 1991; *Ansmann et al.*, 2003; *Groß et al.*, 2012; *Rolf et al.*, 2012; *Kienast-Sjögren et al.*, 2016; *Urbanek et al.*, 2018). The basic product of a polarization lidar is the volume linear depolarization ratio $\delta$ which is defined as the ratio of the returning light power from polarization components perpendicular (cross-polarized) and parallel (co-polarized) to the polarization direction of the transmitted laser source. It includes the scattering of molecules and particles and is thus dependent on particle concentration. In contrast, the particle linear depolarization ratio, defined as the ratio of the perpendicular and parallel component of the backscatter coefficient, characterizes only the scattering properties of particles. It is independent from their concentration and can be used to characterize differences in particle properties. PLDR is a key parameter that is commonly used in the lidar field to quantify the changes in polarization and to retrieve information on ice habit in clouds (e.g. Sassen and Zhu, 2009). Using the measurements of an airborne lidar during the ML-CIRRUS campaign 2014 over Europe (*Voigt et al.*, 2017), *Urbanek et al.* (2018) found enhanced values of the PLDR of cirrus clouds forming in areas of high aviation emissions. They interpreted these changes as an effect of more frequent heterogeneous freezing on aviation exhaust particles. It has long been known that aircraft-emitted particles may act as efficient ice nuclei leading to heterogeneous nucleation in regions with a favorable atmospheric state (including temperature and humidity) (e.g. *Schumann*, 1996; *Jensen and Toon*, 1997; *Kärcher*, 2007). Further, aviation-induced aerosols and contrails can alter the properties of cirrus clouds (e.g. *Tesche et al.*, 2016; *Kärcher*, 2017; *Urbanek et al.*, 2018).

During the COVID-19 pandemic aviation was significantly reduced over Europe. Eurocontrol reports a drop of more than 80% beginning mid of March 2020, with its peak of -88% in April 2020 (Source: www.eurocontrol.int/covid19, last access: 25 June 2021). In May/June 2020 aviation shows a slight recovery to about 40-50% of air traffic compared to the year before. Thus, this episode provides a unique testbed to investigate changes in cirrus cloud properties and occurrence due to reduced
aviation. In our study we use spaceborne lidar measurements from the CALIPSO satellite (*Winker et al.*, 2010) to study cirrus cloud properties over the European region. We focus on this area as an impact on cirrus cloud properties from aviation induced aerosols was found for this region (*Urbanek et al.*, 2018).

Recently, *Schumann et al.* (2021) investigated the induced contrail changes by the air traffic reduction during COVID-19 within the same region by performing contrail simulations with the contrail prediction model of CoCiP (*Schumann*, 2012). They
quantified air traffic and contrail changes from March to August 2020 accordingly and compared them to the same period in 2019. They found that the reduced contrail length for this 6-month period in 2020 was caused only partly by air traffic reduction and partly by less favorable meteorological conditions. Their findings from the model predictions were further estimated by comparison to satellite observations in a parallel paper (*Schumann et al.*, 2021), reaching to a general agreement between observations and modeled data. To largely exclude the effect of meteorological conditions on cirrus occurrence and cirrus
properties in our study, we extended this study to a larger number of years but focused only on March and April measurements as they showed the least differences for the different years and the strongest reduction in air traffic.

In Section 2 we will outline the CALIPSO data and methods. Section 3 describes our results concerning changes of cirrus cloud properties and occurrence in March and April 2020 compared to the previous 6 years (2014–2019). A discussion of our findings, including a significance test, is given in Section 4. And finally, Section 5 concludes this work.

## 2   Data and Methods

The Cloud-Aerosol Lidar and Infrared Pathfinder Satellite Observation (CALIPSO) satellite was launched on April 28, 2006 and is flying as part of the NASA Afternoon Constellation or A-Train in a sun-synchronous polar orbit at an altitude of 705 km with an equator-crossing time of about 1:30 PM and a 16-day repeat cycle (*Winker et al.*, 2010; *Stephens et al.*, 2018). Since September 2018, CALIPSO has moved to a lower orbit (16.5 km lower than the A-Train) to join the CloudSat satellite
in orbit to simultaneously probe the Earth system. The main objectives of CALIPSO mission are to provide information on the vertical distributions of aerosols and clouds as well as their physical properties over the globe with unprecedented spatial resolution which is beneficial to complement current measurements and to improve our understanding of weather and climate. The Cloud-Aerosol Lidar with Orthogonal Polarization (CALIOP) instrument is the primary payload along with an Imaging Infrared Radiometer (IIR) and a Wide-Field Camera (WFC) carried on the CALIPSO satellite. CALIOP is a dual-wavelength
polarization lidar system with a three-channel receiver, optimized for global profiling of aerosols and clouds and their optical and microphysical properties. CALIOP is built around a diode-pumped Nd:YAG laser which produces simultaneous co-aligned pulses at 532 nm and 1064 nm (*Winker et al.*, 2007; *Hunt et al.*, 2009). Each laser produces 110 mJ energy at each of the two wavelengths per pulse with a repetition rate of 20.16 Hz (corresponding to a horizontal resolution of 333 m on the Earth's

surface). The angular divergence of each laser beam is reduced to approximately 100 $\mu$rad thanks to the beam expander on each laser, which results to a footprint of 70 m diameter on the Earth's surface. Backscattered signals are received by a 1-m telescope, which feeds a three-channel receiver. The 1064-nm receiver channel is polarization insensitive and only measures the elastic backscatter intensity. While two polarization-sensitive 532-nm receiver channels independently measure two orthogonal

polarization components which are polarized parallel and perpendicular to the polarization plane of the transmitted beam. Since the launch of CALIPSO, numerous validation studies have been carried out with ground-based (e.g. *Pappalardo et al.*, 2010; *Mamouri et al.*, 2009; *Lopes et al.*, 2013) and airborne (e.g. *McGill et al.*, 2007; *Burton et al.*, 2013) lidar measurements.

    The CALIPSO data used in this study are the Level 2 5-km Cloud Profile Products which contain the information of scientific parameters such as particle linear depolarization ratio, temperature (derived from the GEOS-5 data), ice water content (derived

from the CALIOP retrieved extinction by ice cloud particles) etc. The CALIOP data are stored as half orbits from north to south and thereby separated by day and night. Daytime observations are affected by solar background illumination that decreases the signal-to-noise ratio, making the daytime measurements more challenging to interpret. However, in the north Atlantic flight corridor covering the European region which we are interested in, there is an aviation fingerprint with two maxima during morning eastbound and afternoon westbound traffic (e.g. *Graf et al.*, 2012; *Schumann and Graf*, 2013). In the current study,

therefore, all measurements (both daytime and nighttime) will be analyzed in order to study the influence of air traffic on cirrus clouds to the fullest extent.

    CALIOP has a fundamental sampling resolution of 30 m vertical and 335 m (1/3 km) horizontal, depending on the receiver electrical bandwidth and the laser pulse repetition rate. However, the spatial scales of atmospheric variability tend to increase with altitudes and the backscattered signals from particles (such as clouds and aerosol layers) above that from ambient air

molecules become weaker. To overcome this situation, CALIOP has conducted different averaging algorithms for different altitudes for a better detection of occurring features in the atmosphere (*Vaughan et al.*, 2009; *Winker et al.*, 2009), which allows to retain the fundamental vertical resolution of 30 m in the lower troposphere and to identify the fainter features with required signal-to-noise ratio in the high altitudes. The details of the spatial resolutions of the CALIOP data are listed in Table 1.

    The fundamental measurements made by CALIOP are calibrated altitude-resolved profiles of backscatter intensity from a

variety of geophysical entities, including clouds, aerosol layers, regions of clear air, and the returns from the Earth's surface. Retrievals of aerosol and cloud properties and the correct interpretation of their measurements require first the accurate discrimination between aerosols and clouds within the observed profiles. Furthermore, cloudiness consisting of a variety of cloud types are characterized by different optical and physical properties and have different influence on radiative forcing and precipitation. The CALIPSO team developed the vertical feature mask (VFM) to classify aerosols and clouds based on statistical differences

in the various optical and physical properties of the detected layers and further to separate them into different subtypes (e.g. *Liu et al.*, 2004, 2009; *Hu et al.*, 2009; *Omar et al.*, 2009; *Vaughan et al.*, 2009). The VFM products stored also in the Level 2 data are used in this study to distinguish cirrus clouds from aerosols and non-cirrus clouds.

    A cloud layer product of CALIPSO includes cloud different properties: e.g. cloud height, backscatter, extinction, ice/water phase. In order to exclude misclassified mix-phased clouds and noise-contaminated signals, we only consider measurements

at temperatures below -38 °C (=235 K), above 6 km altitudes and with cloud thickness larger than 0.1 km. The observations

**Table 1.** Spatial resolution of downlinked data from CALIOP at 532 nm

| Altitude range (km) | Horizontl resolution (km) | Vertical resolution (m) |
|---|---|---|
| 30.1–40.0 | 5.025 | 300 |
| 20.2–30.1 | 1.675 | 180 |
| 8.2–20.2 | 1.005 | 60 |
| -0.5–8.2 | 0.335 | 30 |

of cirrus clouds with CALIPSO are used to infer cirrus occurrence rates (OR). This analysis is carried out on single cirrus cloud profiles (determined with VFM) grouping the cirrus clouds in geometrical thicknesses of 100 m, 300 m, 1 km, and 2 km, respectively. The cirrus OR are hence calculated as the ratio of the number of profiles with cirrus cloud layers to the total number of observed profiles. In order to compare the changes of cirrus occurrence and properties under the conditions of reduced air traffic, we consider statistical values of the cirrus OR (here monthly mean) rather instead of single cases.

It is mentioned above that CALIPSO provides global profiling of clouds in the troposphere and lower stratosphere. In this study, however, we have concentrated on the similar area as the ML-CIRRUS campaign (*Voigt et al.*, 2017), more precisely, the whole range of the midlatitudes from 35°N to 60°N and from the Atlantic Ocean (15°W) to central Europe (15°E) (for the sake of simplicity, we call the here-considered area as Europe in the rest of this manuscript). As CALIOP is a nadir-pointing lidar, data is collected only along the ground track of the CALIPSO satellite. CALIPSO flies 3-4 times each day over this area and therefore ∼100 tracks of observations each month were collected in March and April. Further, this area covers a large fraction of the North Atlantic flight corridor connecting central Europe with north America where the generation of contrail-induced cirrus clouds and the aviation impact on cirrus clouds have been intensively studied (e.g. *Graf et al.*, 2012; *Schumann and Graf*, 2013; *Voigt et al.*, 2017; *Urbanek et al.*, 2018; *Schumann et al.*, 2021).

## 3 Results

Cirrus ice crystals generally form in regions of ascending motions (producing the necessary supersaturation over ice) by ice nucleation on aerosol particles in the upper troposphere (in-situ origin cirrus), or they appear in the cloud outflow of frontal systems or convection as frozen cloud droplets that had formed at lower altitudes and warmer temperatures (liquid origin cirrus). Aircraft flying in cold and humid air masses may trigger the formation of contrails by mixing the aircraft exhaust and the surrounding air with water content condensing on the airborne aerosols that might also be emitted by aircraft. After formation, contrails can further spread out as persistent contrails and develop into contrail cirrus clouds when the background air is supersaturated with respect to ice. In addition, the appearing contrail or contrail-cirrus ice crystals might also change the optical properties of naturally occurring cirrus clouds. To exclude that changes found in ice cloud occurrence and properties are caused by substantial differences in meteorological conditions we analyze monthly climate composites of geopotential height (GPH) at 500 mb (as measure for the general circulation pattern) for the region covering the extra-tropical North Atlantic

and the European mainland. For this analysis National Center for Environmental Predictions/National Center for Atmospheric Research (NCEP/NCAR) Reanalysis 1 (e.g. *Kalnay et al.*, 1996; *Kistler et al.*, 2001) are applied. The corresponding data can be achieved through Physical Sciences Laboratory, NOAA, Boulder, Colorado, from their website at https://psl.noaa.gov/. We found a general good agreement for the circulation patterns in March (see GPH at 500 mb in Figure 1), especially for the comparison between 2020 and the composite mean of the previous 6 years from 2014 to 2019. Looking at the year-to-year variability 2016 and 2018 showed slight differences with a stronger component of northwesterly flow in the western part of our observation area for 2016 and an eastward shift of the general circulation pattern in the observation area for 2018. The circulation patterns in April (see Figure 2), however, showed more variabilities between different years than in March. The comparisons fortunately showed that the results in 2020 were not an outlier from others, but fell well within the spread of the variabilities in the considered years. Besides the GPH at 500 mb, we further compare the general meteorological conditions along the entire altitude range covering our observations in terms of mean profiles of temperature, relative humidity with respect to ice (RHi), and vertical velocity over our research area in the years 2014–2020 in March and April, respectively. These parameters are directly derived from global ERA5 reanalysis data, produced by ECMWF within the Copernicus Climate Change Service (*Hersbach et al.*, 2020) and the results are shown in Figure 3. Looking at the year-to-year variability in March we note that the profiles of temperature and RHi show departures in 2018 and 2020 compared with nearly identical values in the other years. The temperatures in 2018 and 2020 are on average nearly 4 °C higher than in other years at altitudes above ∼10 km, whereas below 10 km temperatures are nearly identical in 2020 but slightly lower in 2018 compared with other years. The RHi profile in 2020 shows lower values along the entire altitudes, while the results in 2018 are lower at higher altitudes above ∼10 km and higher at lower altitudes compared with other years. In April, however, the profiles of temperature show nearly identical values at lower altitudes below ∼10 km and a larger spread above 10 km with the 2020 results falling within the spread. The RHi values in 2020 are comparable with the results in the previous years (slightly drier in the lower altitudes below 10 km). In addition, warmer and drier airmasses were found in 2016 and 2018, especially at the altitudes between ∼9 and 12 km. Finally, the profiles of vertical velocity in each year are quite different from each other in both March and April. The vertical velocities vary on average within the range between -0.3 and 0.3 cm/s. It is important to note that the 2020 results fell within this spread of vertical velocity. Furthermore, the large variabilities in the mean profiles of vertical velocity did not lead to big difference in the occurrence rate of cirrus clouds, which will be shown below. In addition, looking at the year to year variability for the time period May-August we conclude that the general meteorological conditions might have a quite large impact on weather and cloudiness in the observation area. Thus we confine our study to March and April data. With a general picture of meteorological conditions in mind, we use CALIPSO data of March and April to investigate changes in the cirrus cloud occurrence and properties caused by reduced aviation.

## 3.1 Geometrical thickness and occurrence rate

We first compare the geometrical thickness of cirrus clouds which is defined as the vertical extension of cirrus clouds; no matter how many layers the clouds can be characterized by (i.e., either the clouds are continually distributed or not). In the analysis, cirrus clouds with thickness smaller than 0.1 km are considered as cirrus free and the corresponding observations

will be neglected. The calculated occurrence frequencies of the cirrus thicknesses from the observations in March are shown in the histograms (bar width of 0.2 km) in Figure 4. From all the here-analyzed observations, the distributions of the cirrus thicknesses are positively-skewed with a long tail extending to larger values up to ∼5 km. There are maximum occurrence frequencies found to fall within the range of 0.1–1.5 km. The decrease of occurrence frequencies of cloud thickness towards larger values is much sharper for the results in 2020 than in the previous years. It means that there were much less thick clouds occurring in March 2020. The calculated average thicknesses vary from 1.31 to 1.43 in the years from 2014 to 2019 with a 6-year mean of 1.38 km which are in good agreement with the typical value of cirrus thickness of 1.5 km reported by previous studies (e.g. *Dowling and Radke*, 1990), although the use of the temperature limit of -38 °C may reduce the actual cirrus geometrical thicknesses. The average thickness of cirrus clouds in March 2020, however, is much smaller and significantly reduced to only 1.18 km.

The geometrical thicknesses in April of the years 2014–2019 are in general very close to, or slightly smaller than, the results in March. The 2020 results in April, however, seem to recover from March and become close to the previous years with only 70 m less extent than the 6-year mean of 1.36 km. It is compared above (see Figure 3) that the meteorological conditions of March in 2020 were less favorable for cirrus clouds to form and maintain than in the previous years and the reduction in cirrus thicknesses in March was too strong to be only due to the aviation reduction. While the scenario in April was different and will be further discussed below.

We next show the cirrus occurrence rate (OR) in April for the different years in Figure 6. Please note that we will only show the resulting cirrus OR and particle linear depolarization ratio (in the next subsection) in April, since the results in March and April provide us the same information. However, the medians of the corresponding parameters in March will be summarized in Tables 2 and 3. For all the years considered in this study the OR profiles show the maxima at the altitude of about 9.5 km (9.0 km for 2018). However, cirrus clouds in 2016 and 2020 show a reduced OR of only about 9–10% compared to about 12% in other years. A clear reduction of cirrus OR in April 2016 as well as in 2018 but only for the higher altitude regions is supposed to be due to the meteorological conditions with warmer and drier airmasses as mentioned above. We hence further consider the previous years without 2016 and 2018 as reference years for easy description. In general, cirrus clouds were found in a height range of 6–14 km, where the cirrus OR for 2020 shows a clear reduction in the height range from about 7 to about 12 km compared with the reference years. The profiles of cirrus OR for the reference years show almost no variation. Comparing the profiles of cirrus OR for March (not shown) we found similar conditions also for 2016. I.e., we see no apparent differences of the OR profiles for the years 2014–2017 and 2019 but a clear reduction of OR for 2020. 2018 (March results) shows the behavior similar to April with a reduction only in the uppermost altitude ranges compared to all other years. To further explore the reason leading to the reduction of cirrus OR, we divide the data into a subset for temperatures from -50 to -38 °C and colder than -50 °C. The resulting cirrus OR within different temperature range show cirrus clouds occurred at altitudes from about 8 to 14 km at colder temperatures and from 6 to 11 km at warmer temperatures (see the middle panel of Figure 6). The corresponding maxima of cirrus OR are found at ∼8 and 10.5 km at different temperatures, respectively. It is well known that T = -50 °C is one of the threshold conditions for contrail formation (*Schumann*, 1996). Strong aviation reduction in April 2020 will lead to the reduction of contrail formation which further influences the cirrus cloud coverage (e.g. *Schumann et al.*, 2021).

**Table 2.** The occurrence rates of cirrus clouds with the definition based on different geometrical cloud thicknesses larger than 0.1, 0.2, 1.0, and 2.0 km, respectively, in March of years 2014–2020.

| Year (March) | Cirrus occurrence rate (%) | | | |
|---|---|---|---|---|
| | > 0.1 km | > 0.3 km | > 1.0 km | > 2.0 km |
| 2014 | 31.0 | 28.7 | 17.2 | 6.8 |
| 2015 | 32.4 | 29.9 | 17.2 | 6.1 |
| 2016 | 32.1 | 29.7 | 17.3 | 6.3 |
| 2017 | 37.8 | 34.8 | 20.9 | 7.8 |
| 2018 | 35.3 | 31.6 | 16.8 | 5.2 |
| 2019 | 31.1 | 28.9 | 16.5 | 6.3 |
| 2020 | 25.7 | 22.9 | 11.7 | 3.5 |

Indeed, the cirrus OR in 2020 showed a clear reduction at temperatures below -50 °C from 8 to 12 km by up to 3% (such as 8% in 2020 against 11% in 2017 at the maxima) compared with the reference years. At warmer temperatures, however, the cirrus OR in 2020 were smaller than in the reference years at lower altitudes below 9 km and became close to and even slightly larger than the other years at higher altitudes. The cirrus OR depends on the geometrical thickness of the cloud (see the right panel of

Figure 6). The largest reduction in cirrus OR in 2020 is found for geometrical thicknesses larger than 0.1 km, 0.3 km and 1.0 km, with an OR of 25%, 23%, 13%, respectively. The reference years show values of >31%, >28%, and > 17%, respectively. Please note that the thinner cirrus clouds are more likely to originate from the contributions of contrails and contrail cirrus than the thicker cirrus clouds which are connected with the convections. As expected, the outlier is 2016 which was characterized with a cirrus OR slightly larger than 2020 but much smaller than the other years. The cirrus OR for a geometrical thickness

> 2.0 km shows reduction of overall about 5% in 2020 compared to more than 6% of the reference years in 2015, 2017, and 2019 and almost no variations compared with 2014, 2016, and 2018. From the current analysis, it is striking to note that the cirrus OR in April 2020 are smaller by a factor of 17–30% than the values derived in the reference years in despite of the cloud thicknesses on which a cirrus cloud was defined. The same findings, although with a smaller proportion, are also seen in the observations of March 2020 (see Table 2). Our results are consistent with the previous findings that air traffic might increase

the occurrence of cirrus clouds (*Boucher*, 1999).

### 3.2 Cirrus particle linear depolarization ratio

We next compare the relation between the cirrus PLDR with the corresponding ambient temperatures in different years. The temperatures used for this comparison are derived from the GEOS-5 (Goddard Earth Observing System, Version 5) model data product provided to the CALIPSO by the GMAO data assimilation system. The determined relations are shown in Figure 7

where a heatmap coloring is used to specify the relative number density of the scatter point data with the maximum number density indicated by 1 in the corresponding colorbar. First of all, there are hotspots (with a large amount of data points) found for all cases at the temperatures higher than ∼-45 °C where PLDR mostly fall into a range between ∼0.20 and 0.50. This hotspot

is similar for all the analyzed years. We further note that there is a secondary hotspot within the temperature range of -60 °C to -50 °C with larger PLDR of up to 0.60 in the previous years including 2016. Less cirrus clouds, however, were detected at this lower temperature range (<-50 °C) in 2020. In addition, at temperatures higher than -50 °C there are no clear correlations found between PLDR and temperatures. At -50 °C and colder, however, there is a clear negative correlation between PLDR and

temperatures, namely that cirrus PLDR increase with falling temperatures which agrees with many previous cirrus observations (e.g. *Sassen and Benson*, 2001; *Urbanek et al.*, 2018).

In order to further clarify this feature, we divide the data into a subset for temperatures from -50 to -38 °C and colder than -50 °C. Before going into details, it is important to mention that the PLDR values below 0.10 and above 0.80 were cut off, for the consideration that those values are correlated with large uncertainties and should be unphysical. Resulting

histograms of cirrus PLDR and their median values for the different temperature regimes are shown in Figure 8. First of all, the histograms of cirrus PLDR can be characterized by a right-skewed distribution with a long tail extending to larger values. The distributions of PLDR show that in general the PLDR values at lower temperatures (<-50 °C) are larger than at higher temperatures (>-50 °C), namely that the distributions of PLDR at lower temperatures have a larger skewness to the right (larger values). Focusing in more details on the comparisons, however, we note that the distributions of PLDR at higher temperatures

are in good agreement for all the cases in years 2014–2019 with median values of 0.342 for the 6-year composite and a slightly smaller median of 0.330 in 2020. The median values in different years are indicated on the corresponding panels. While the PLDR at temperatures colder than -50 °C show a significant reduction in 2020 compared to the previous years including 2016. The medians of cirrus PLDR for the years 2014–2017 and 2019 are quite similar with medians varying from 0.390 to 0.394, respectively. It is described above that the vertical distribution of the cirrus occurrence in 2018 was shifted downwards by ∼0.5

km compared with other years. Hence there might be more cirrus clouds in 2018 occurring outside of the aviation cruising altitudes than other years, which leads to the lower PLDR values with a median of 0.378. The median of the cirrus PLDR for 2020, however, is only 0.360 at temperatures <-50 °C. From the March results, we also see the same feature with nearly identical medians of PLDR at warmer temperatures and a reduction in PLDR medians in 2020 compared with the previous years (see Table 3). The possible interpretation for reduced PLDR found in March 2018 at colder temperatures is the same as

for the results in April 2018. As, besides the dependence on the temperature, the PLDR might also depend on an aviation effect, it should also be visible for the different meteorological conditions. This feature of PLDR can be interpreted by the fact that the contrails which may lead to contrail-induced cirrus characterized by higher PLDR were observed at temperatures below around -50°C (e.g. *Schumann*, 1996; *Voigt et al.*, 2011) in the normal years, whereas lack of contrails due to the reduction of air traffic in April 2020. In addition, we note that the cirrus clouds in April 2016 were characterized by significantly reduced

occurrence rates due to warmer and drier airmasses, while their PLDR were comparable to the other reference years.

We also compare the vertical profiles of the PLDR median (Left panel in Figure 9 - solid lines) along with the corresponding 25th and 75th percentiles (dashed lines) for the whole height range between 6 and 15 km. The resulting profiles of PLDR show a well-known increase with increasing altitudes (e.g. *Urbanek et al.*, 2018) for all the cases at the typical aviation cruising altitudes between 8 and 12 km. However, the median values of the cirrus cloud's PLDR profile in 2020 were reduced to only

about 0.31 at 8.5 km and about 0.38 at 11.5 km compared to the medians from about 0.34 at 8.5 km to 0.41 at 11.5 km for

**Table 3.** Medians of the cirrus particle linear depolarization ratio determined at warmer (-50 °C < T < -38 °C) and colder temperatures ( T < -50 °C), respectively, in March of years 2014–2020.

| Year (March) | Cirrus particle linear depolarization ratio | |
| --- | --- | --- |
| | -50 °C < T < -38 °C | T < -50 °C |
| 2014 | 0.344 | 0.401 |
| 2015 | 0.347 | 0.404 |
| 2016 | 0.347 | 0.391 |
| 2017 | 0.349 | 0.401 |
| 2018 | 0.344 | 0.373 |
| 2019 | 0.346 | 0.396 |
| 2020 | 0.344 | 0.377 |
| 2014-2019 | 0.346 | 0.395 |

the composite means of the previous years. Besides these reduced values the profile of the April 2020 data showed the same behavior (altitude dependence) as the previous years. At altitudes below 8 km and above 12 km, however, there are larger variabilities in PLDR due to the lower occurrence of cirrus clouds and no clear altitude dependence found. Furthermore, we also show the corresponding ambient temperatures as well as the temperatures in cirrus clouds in different years in the right panel of Figure 9. The air temperatures show a slightly larger spread at higher altitudes (above ∼10.5 km) and the 2020 results within the spread, the same information as shown in Figure 3 (lower left panel). The temperatures in cirrus clouds, however, show a nearly same spread along the altitudes. Only exception was found for 2020 above ∼11.8 km with a much colder temperatures, which implies the formation of ice crystals at lower temperatures due to less ice nucleation particles. The similar feature was found for the March measurements, however, with the decreases of the PLDR with height only found at altitudes higher than ∼10 km in 2020 (not shown here). We should mention that the reduction of air traffic over Europe started from the beginning of March, e.g., air traffic over Germany reduced to about 40% on March 17 and to 80% on March 25, and remained in entire April (Source: www.eurocontrol.int/covid19, last access: 25 June 2021).

## 4 Discussion

Our analysis above shows that comprehensible and precise reductions were found in the occurrence rates and thicknesses of cirrus clouds during the period of coronavirus pandemic in March and April, 2020, when the public air traffic was significantly reduced (more than 80% in entire April). Before we may draw final conclusions on the findings, a significance test and parallel comparisons with other regions will be further carried out.

It is mentioned above that the derived PLDR of cirrus clouds are not normally distributed. In order to test the significance of difference between the cirrus PLDR in different years, we here applied a Mann-Whitney U test which is a widely-used nonparametric test for equality of variable medians of two independent samples. Before taking the exercise, we have to down-

sample the data since the datasets have a huge number of data points. The sampling has been done for a function varying in time at the same altitude with a sampling rate of 1/10, i.e., one data point was sampled from every 10 points, although the data set after sampling still have more than 20 thousand data points. It was shown above that air traffic mainly exerts influence on the distributions of PLDR at lower temperatures (< -50 °C) by inducing the formation of contrails and contrail cirrus. Exhaust soot particles, however, also cause indirect effects on naturally occurring cirrus by increasing heterogeneous nucleation (e.g. *Urbanek et al.*, 2018). We hence focus on the observations at altitudes between 8 and 13 km, which are the typical cruising altitudes for passenger and cargo aircrafts. In addition, we will carry out the significance test by randomly choosing the reference years of 2014, 2017, and 2019. Comparisons between the sampled data and the corresponding original data in different years have also been done, respectively, showing the same (or similar) distributions with a high significance level. The overall results of the Mann-Whitney U test at a significance level of $p = 5\%$ are presented in Tables 4 and 5. Here, $p$-value returned from a Mann-Whitney U test is a measure of the probability to reject or retain the null hypothesis, i.e., the two samples follow continuous distribution with equal medians. $h$ is a logical value (0 or 1) to give the test decision: $h = 1$ indicates a rejection of the null hypothesis and $h = 0$ indicates a failure to reject it at the $5\%$ significance level. For the observations in March, it is striking that the distributions of cirrus PLDR in the years of 2014, 2017, and 2019 are significantly the same (with $p > 5\%$). While the resulting PLDR in 2020 are significantly different from the previous years ($p = 0$). In April, we see the 2020 results are again significantly different from the reference years and the PLDR distributions in 2017 and 2019 are nearly the same ($p = 95.6\%$). However, the distributions of PLDR in 2014 are slightly different from the results in 2017 and 2019, although the $p$-values are larger than $1\%$.

**Table 4.** Significance test using Mann-Whitney U test: March, Europe

| year | 2014 | 2017 | 2019 | 2020 |
|------|------|------|------|------|
| 2014 | $p = 1, h = 0$ | $p = 0.3356, h = 0$ | $p = 0.3537, h = 0$ | $p = 5.46e - 21, h = 1$ |
| 2017 | | $p = 1, h = 0$ | $p = 0.0618, h = 0$ | $p = 4.10e - 26, h = 1$ |
| 2019 | | | $p = 1, h = 0$ | $p = 2.48e - 17, h = 1$ |
| 2020 | | | | $p = 1, h = 0$ |

**Table 5.** Significance test using Mann-Whitney U test: April, Europe

| year | 2014 | 2017 | 2019 | 2020 |
|------|------|------|------|------|
| 2014 | $p = 1, h = 0$ | $p = 0.0244, h = 1$ | $p = 0.0118, h = 1$ | $p = 2.34e - 82, h = 1$ |
| 2017 | | $p = 1, h = 0$ | $p = 0.9560, h = 0$ | $p = 7.58e - 69, h = 1$ |
| 2019 | | | $p = 1, h = 0$ | $p = 1.27e - 78, h = 1$ |
| 2020 | | | | $p = 1, h = 0$ |

It is mentioned in the Introduction that the periods of coronavirus pandemic in different regions are different. It provides us a great opportunity to compare the properties of cirrus clouds detected at different regions to study how the reductions of air traffic influence on the cirrus clouds and what the response time of the changes are. For the locations of the study regions we concentrated on Europe (Latitude: 35°N–60°N; Longitude: 15°W–15°E), China (Latitude: 20°N–45°N; Longitude: 90°E–130°E), and the United States of America (Latitude: 30°N–50°N; Longitude: 125°W–75°W), respectively. All the regions are located within the midlatitudes. To determine the regional difference of the occurrence rates of cirrus clouds, we further extend the same analysis to the observations over China and USA. The corresponding results are shown in Figure 10. First of all, the cirrus OR over these three regions show that on average the cirrus clouds occurred more frequently over Europe and USA than over China, although exception was seen in the April results of 2016 over China with an extremely high OR. The findings show overall agreement with previous studies (e.g. *Sassen et al.*, 2008). Furthermore, we see the same changes in the cirrus OR in April over USA as over Europe but no clear changes in March over USA. For China we found no clear changes of cirrus OR in both March and April.

We next turn to compare the cirrus PLDR in different regions. The same pre-analysis as described above were also carried out, namely that only the observations with the PLDR values between 0.10 and 0.80 and at altitudes between 8 and 13 km are considered. The corresponding results over Europe, USA, and China, are shown in Figure 11. The boxes (in black) represent 25th–75th percentiles of all the PLDR values showing the middle 50% of the data (i.e., box bottom stands for the lower quartile and top for the upper quartile). The medians representing the mid-point of the data set are shown by the red lines through the corresponding boxes and the means are shown by the red circles. For all the cases, the PLDR possess a larger mean than the corresponding median, indicating that the distributions of the PLDR are positively skewed as has been reported above. The

box plots provide a general picture of the determined PLDR of cirrus clouds over these three regions. Focusing on the results of Europe, we see the same properties of cirrus clouds as stated above, namely that the cirrus PLDR show excellent agreement with each other in the previous years 2014–2019 whereas reduced values in 2020 in both months. Further, the reduction of the PLDR in March is slightly smaller than that in April, which should be somehow correlated with the different periods with reduced air traffic in March and April over Europe. We next focus on the results observed over USA and China and both showed a slightly larger year-to-year variability than over Europe (of course excluding the 2020 results). Besides this point, we stress that the PLDR values in 2020 show no clear reduction in March but a significant reduction in April over USA, which is expected because the outbreak of coronavirus in USA started a few days later than in Europe and the domestic flights were sharply reduced only in April in USA (Source: www.airlines.org/dataset/ and www.eurocontrol.int/covid19, last access: 25 June 2021). From the observations over China, there are no clear reductions found in 2020 in both months. We would not expect strong changes in cirrus PLDR over China as the aviation effect on cirrus clouds in this region is in general very low (e.g. *Stettler et al.*, 2013; *Righi et al.*, 2021). As for the smaller values of PLDR in April 2014 over China, we checked the altitude profiles of cirrus occurrence and found that most of cirrus clouds in this month occurred at lower altitudes by ∼1 km than other cases (not shown here). This feature is consistent with the results in April 2018 over Europe.

In addition to the different outbreak times of the coronavirus pandemic in different regions, it is important to note that Europe and USA are located more within the aviation corridors than China (e.g. *Stettler et al.*, 2013) and hence the formation and properties of cirrus clouds over Europe and USA are under more impacts by air traffic. The reduction of air traffic during the pandemic may lead to larger changes in cirrus clouds over Europe and USA than over China (e.g. *Righi et al.*, 2021). The cirrus PLDR derived over China show on average smaller values than over Europe and USA with the middle 50% (see the boxes in Figure 11) covering a narrower range of PLDR (∼0.03–0.04 less). While the PLDR of cirrus clouds over the latter two regions are in good agreement with each other. This feature further strengthens the assumption that air traffic does not have a major impact on cirrus cloud coverage and properties over China.

## 5  Conclusions

The abrupt outbreak and rapid spread of the coronavirus disease (COVID-19) pandemic have become a global public health crisis. In order to curb the spread of the pandemic, most, if not all, governments worldwide have carried out containment measures including lockdowns, quarantines, curfews and restriction of public air traffic as well. In the current paper, we have presented the geometrical thicknesses, occurrence rates, and PLDR values of cirrus clouds observed by the space-borne lidar CALIOP on the CALIPSO satellite over the north Atlantic and the European mainland during the period of coronavirus pandemic with reduced public air traffic in 2020. The results have been compared with the corresponding observations in the previous years 2014–2019, not affected by air traffic reduction. Cirrus clouds have been retrieved using the VFM products along with additional filters including a temperature mask ($T < -38$ °C), height threshold ($h > 6$ km), and cloud thickness threshold ($> 0.1$ km).

The geometrical thicknesses of cirrus clouds were first determined and their histograms can be characterized by a right-skewed distribution with maximum occurrence frequencies between 0.1 and ∼1.5 km for March and between 0.1 and ∼2.0 km for April. Furthermore, we notice that there is a much sharper decrease towards larger values in 2020 compared with the previous years in March. The same feature is also seen in April, but with a smaller extent. The calculated average thicknesses from the March data show a much smaller value of only 1.18 km in 2020 compared with approximately 1.40 km in the previous years. For the April data, however, the average thicknesses in 2020 are only slightly smaller than the previous years with only 70 m less extent than the composite mean of 6 years 2014–2019. The determined altitude profiles of the cirrus occurrence rates show that cirrus clouds occurred mostly within the altitude range between 7 and 13 km with the maximum occurrence at ∼9.5 km in the years 2014–2017 and 2019–2020 for both months (∼9 km in 2018). Besides this general agreement, the cirrus occurrence rates show reduced values in 2020 compared with the reference years excluding 2016, especially for the height range of 8–12 km in which most public air traffics take place. The lower cirrus OR detected in April 2016 is supposed to be due to less favorable meteorological conditions with warmer and drier airmasses. The data were further divided into a subset for temperatures between -50 and -38 °C and temperatures below -50 °C. The corresponding cirrus OR show a clear reduction in 2020 at temperatures below -50 °C from 8 to 12 km compared with the previous years excluding 2016 and 2018 which had warmer and drier airmasses and no clear difference at temperatures warmer than -50 °C, especially at altitudes higher above ∼9 km. The same features were also seen in the comparisons of cirrus occurrence rates determined according to different definitions of cirrus clouds as a function of cloud thickness. Namely that the cirrus clouds in April 2020 occurred less frequently by a factor of 17–30% than the corresponding periods in the reference years in despite of the cloud thicknesses used to define a cirrus cloud.

Turning next to a comparison of the particle linear depolarization ratio (PLDR) of cirrus clouds, we divide the data again into a subset for temperatures between -50 and -38 °C and for temperatures below -50 °C, since the correlations between PLDR and the ambient temperatures show different features at different temperatures (see Figure 7). For all the cases, the histograms of cirrus PLDR follow a right-skewed distribution with a long tail extending to larger values other than a Gaussian function. In general, the PLDR values are on average larger at lower temperatures (T < -50 °C) than at higher temperatures. Comparisons between different years show that PLDR values at higher temperatures (-50 °C < T < -38 °C) are nearly the same for all the cases (slightly smaller values for 2020); while PLDR values at lower temperatures (T < -50 °C) are smaller in 2020 than in the previous years at a high significance level. The altitude profiles of PLDR medians as well as the corresponding 25th and 75th percentiles have been further calculated for the whole height range. The results show in general an increase with increasing altitudes for all the cases and PLDR values in April 2020 are nearly parallel along altitudes smaller than in the previous years at the aviation cruising heights between 8 and 12 km. It is reported in literature that aviation leads to the formation of contrails and more frequent occurrence of heterogeneous freezing on aircraft exhaust particles which further leads to the formation of high-PLDR cirrus clouds. Our findings of smaller cirrus cloud occurrence rates and PLDR in 2020 caused by the reduction of air traffic are supported by this scenario.

In order to clarify the influence of air traffic reduction on the properties of cirrus clouds, the observations over China as well as over the United States of America have also been analyzed and compared with the results over Europe. Cirrus

clouds observed over USA show similar properties in terms of occurrence rates and PLDR compared with Europe; while the observations over China are characterized by smaller occurrence rates and PLDR of cirrus clouds compared with the former two regions. Besides the regional discriminations of cirrus clouds, the changes of cirrus cloud properties (PLDR) conform to the timeline of the outbreak of the coronavirus disease and the consequent restriction of air traffic in the here-compared regions.

5    *Acknowledgements.* The authors acknowledge support by the DLR-project MABAK (Innovative Methoden zur Analyse und Bewertung von Veränderungen der Atmosphäre und des Klimasystems). The work for this study has also received support from the POLDIRAD funding. We thank the NASA Langley Research Center Atmospheric Science Data Center (ASDC) and CALIPSO science team for making the data available for research.

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

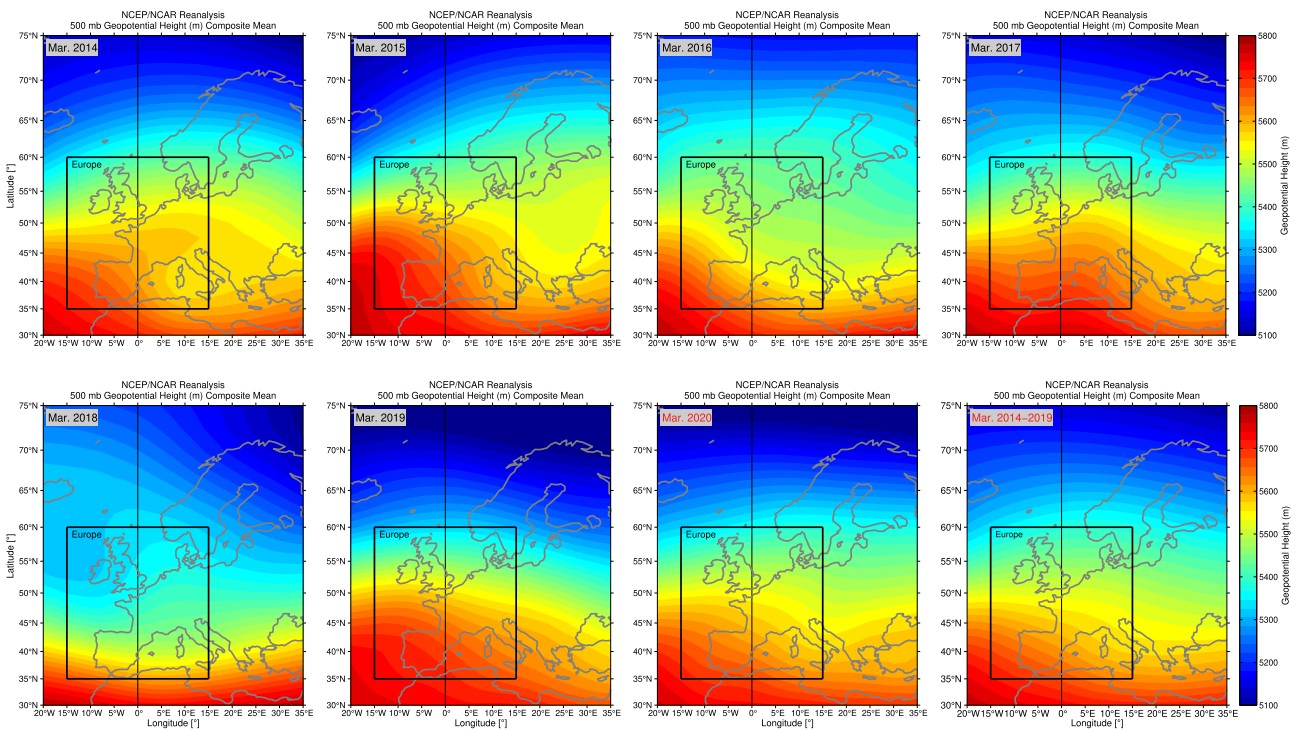

**Figure 1.** 500-mb geopotential height composite mean in March for years from 2014 to 2020 (with years indicated on the plot) and for the average in the previous years 2014–2019 (Rightmost-bottom panel) over Europe. The black boxes indicate the research area of this study. The plots are reproduced based on NCEP/NCAR Reanalysis provided by Physical Sciences Laboratory, NOAA, Boulder, Colorado, from their Web site at https://psl.noaa.gov/.

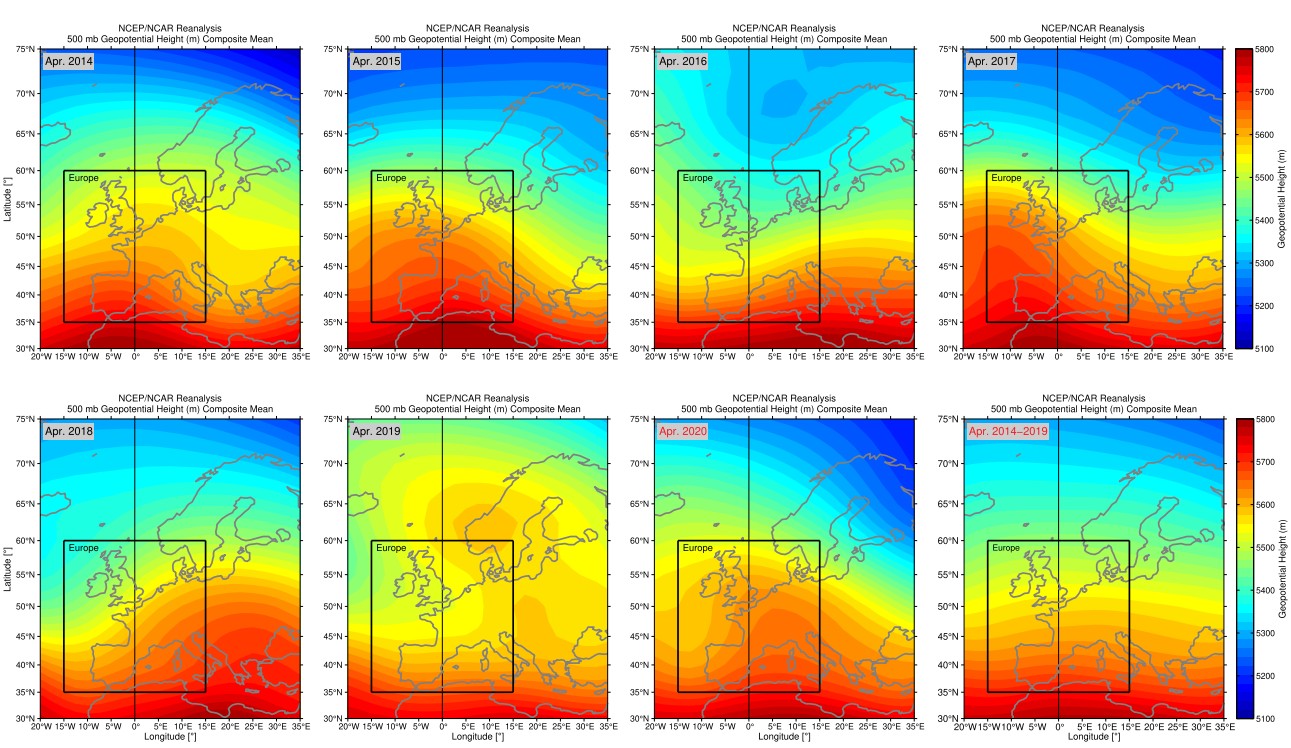

**Figure 2.** Same as Figure 1, but for 500-mb geopotential height composite mean in April.

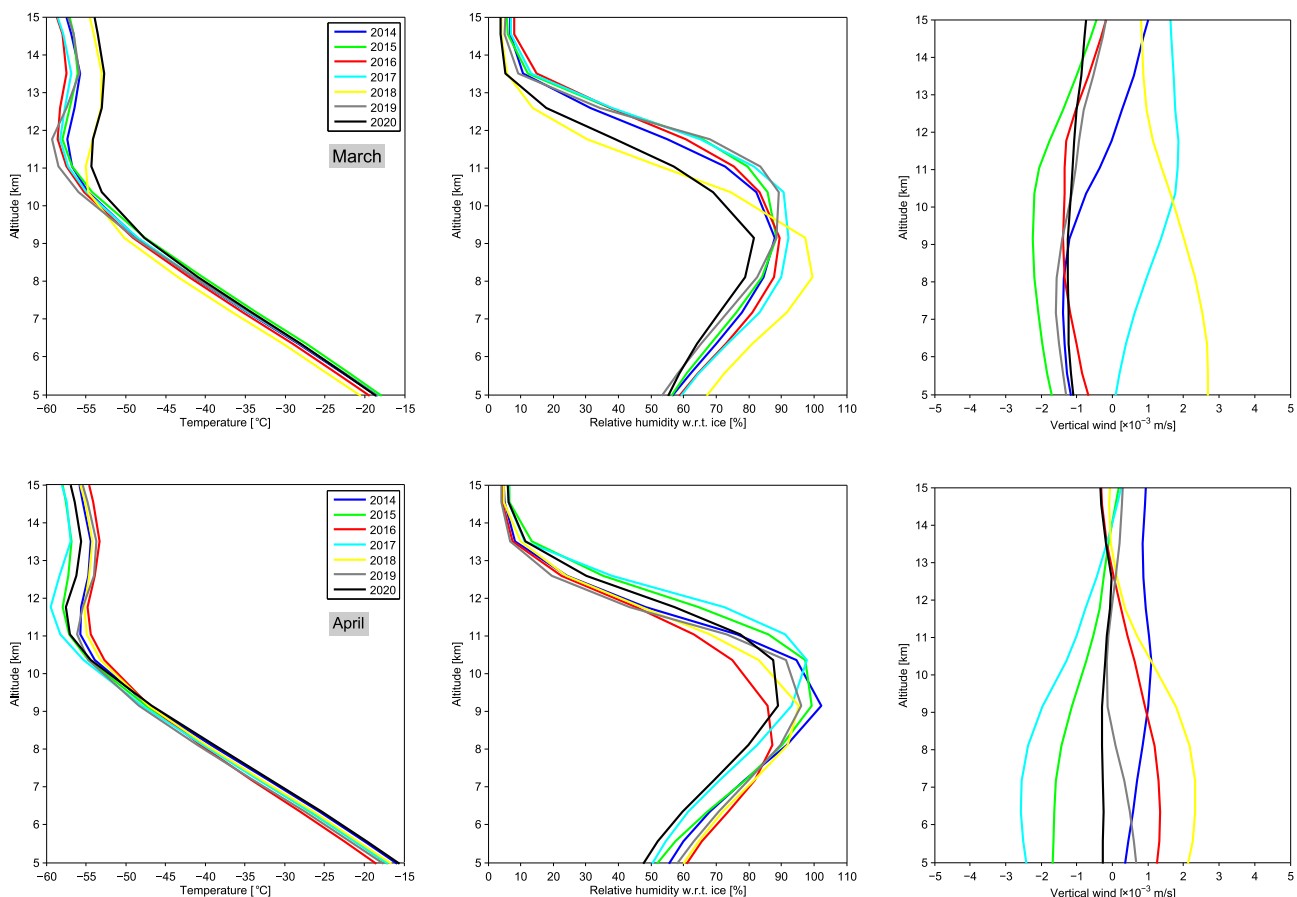

**Figure 3.** Altitude profiles of background temperature, relative humidity with respect to ice (RHi), and vertical updraft derived from ERA5 reanalysis data over Europe (the same area indicated with the black boxes in Figure 1 and 2). Upper panels indicate the meteorological parameters in March and lower panels in April.

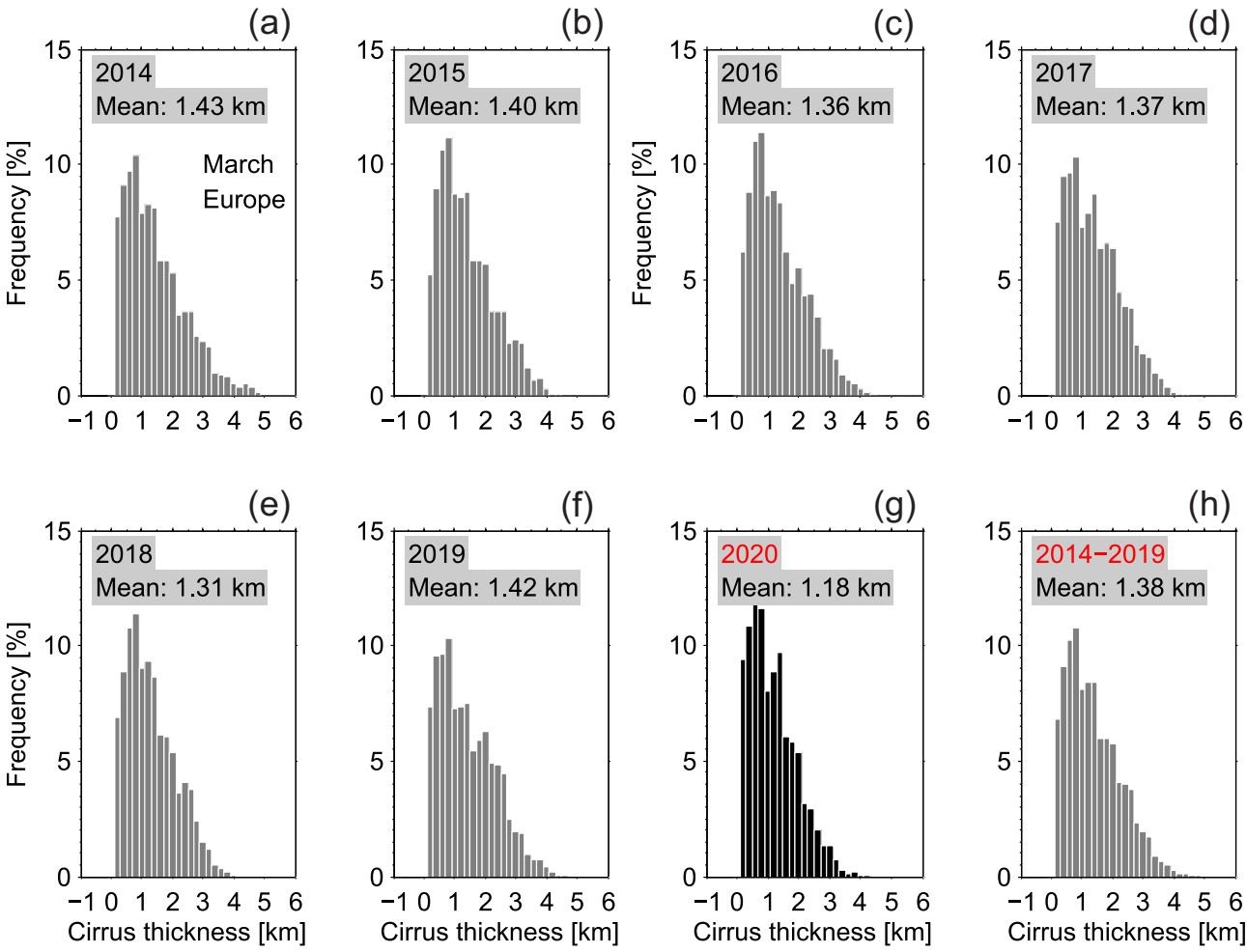

**Figure 4.** Histograms of the geometrical thicknesses of cirrus clouds detected in March in 2020 (shown in Panel (g)) compared with the results in the previous years 2014–2019. The composite results of 6 years 2014–2019 are shown in Panel (h). The average cloud thicknesses for each case are indicated on the corresponding panels.

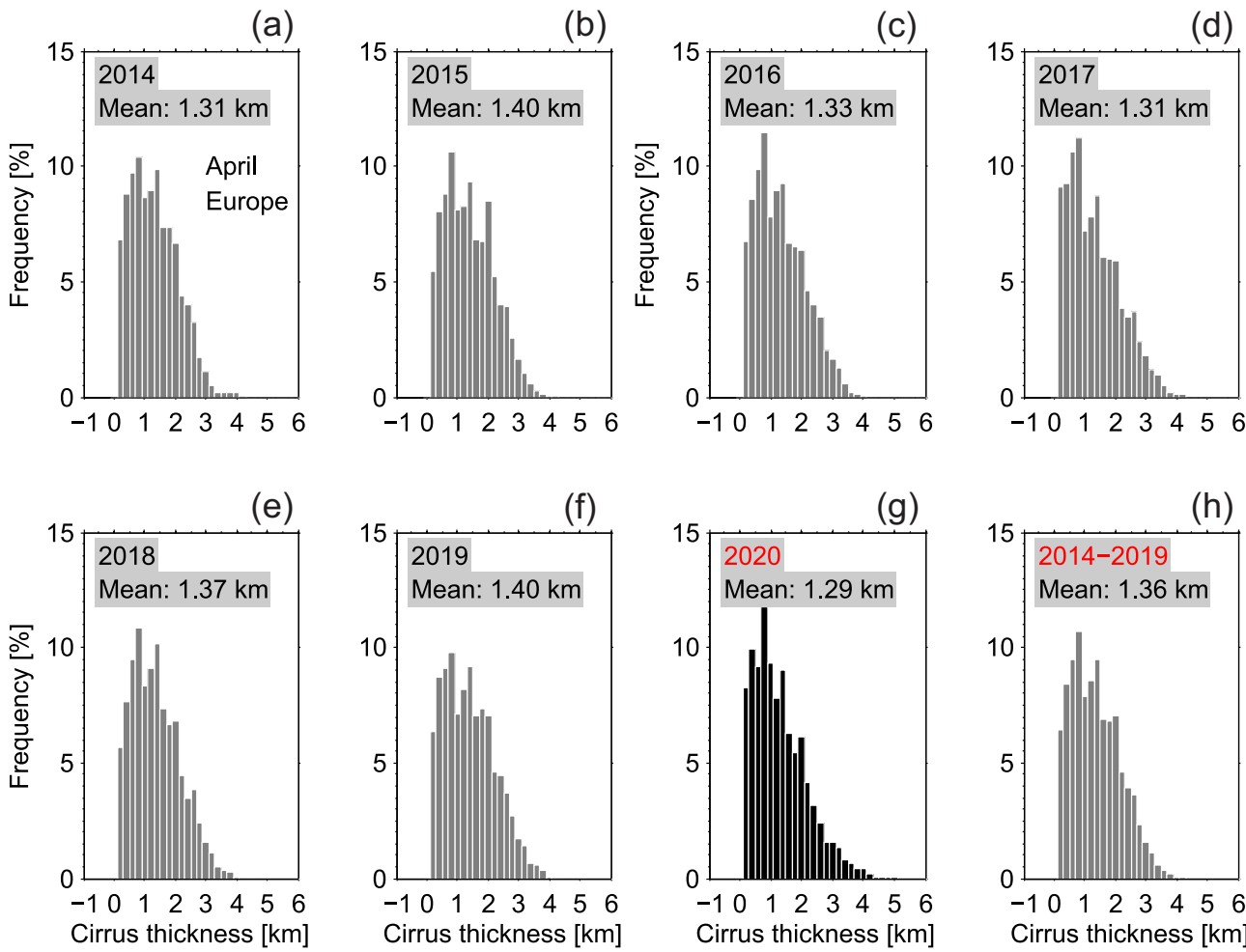

**Figure 5.** Same as Figure 4, but for the geometrical thicknesses of cirrus clouds in April.

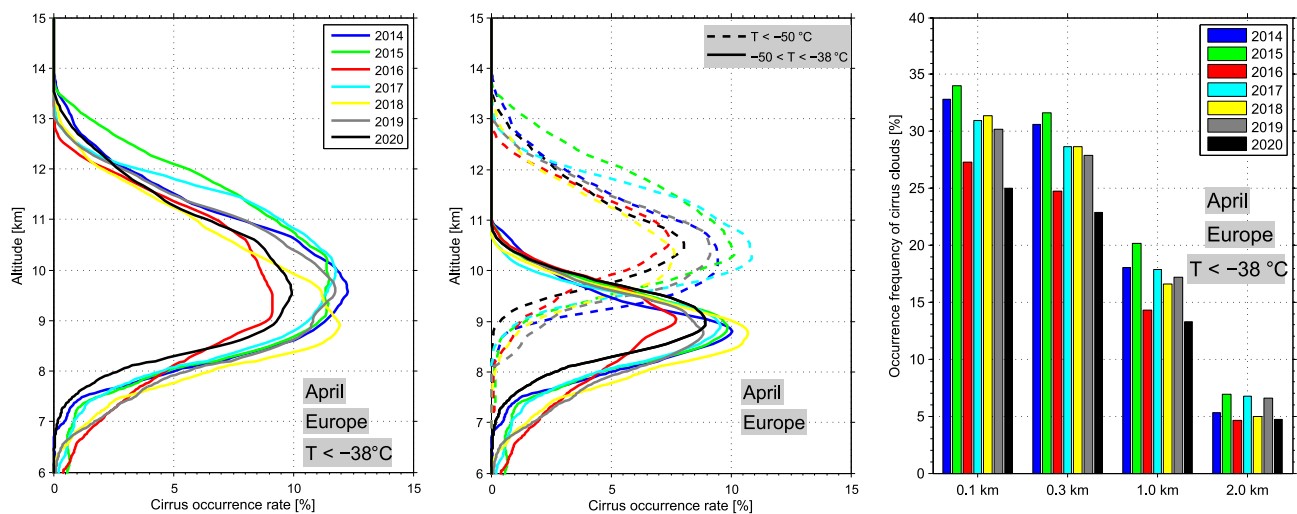

**Figure 6.** Occurrence rates (OR) of cirrus clouds derived from the lidar measurements of CALIPSO in April in different years from 2014 to 2020. Only observations of cirrus clouds at temperatures below -38 °C, at altitudes above 6 km and with cloud thickness larger than 0.1 km are analyzed. Left panel: Altitude profiles of cirrus OR derived from all the observations with T < -38 °C; Middle panel: Altitude profiles of cirrus OR with T < -50 °C (in dashed lines) as well as with -50 < T < -38 °C (in solid lines); Right panel: Histograms of the occurrence frequency according to the definitions of different cloud thicknesses, respectively.

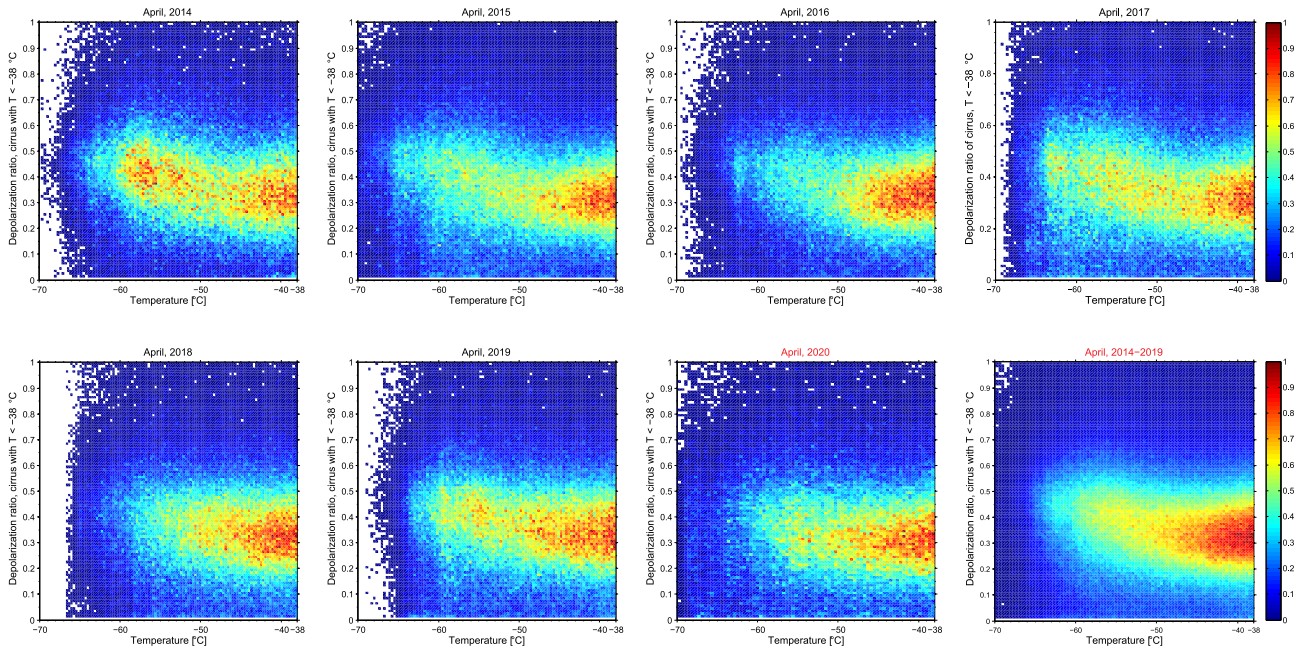

**Figure 7.** Correlations between the particle linear depolarization ratios (PLDR) of cirrus clouds and the ambient temperatures derived from the lidar measurements of CALIPSO and GEOS-5 model data, respectively. The color codes are used to visualize the relative number densities of scatter point data with the maximum number density indicated by 1 in the corresponding colorbar for each case.

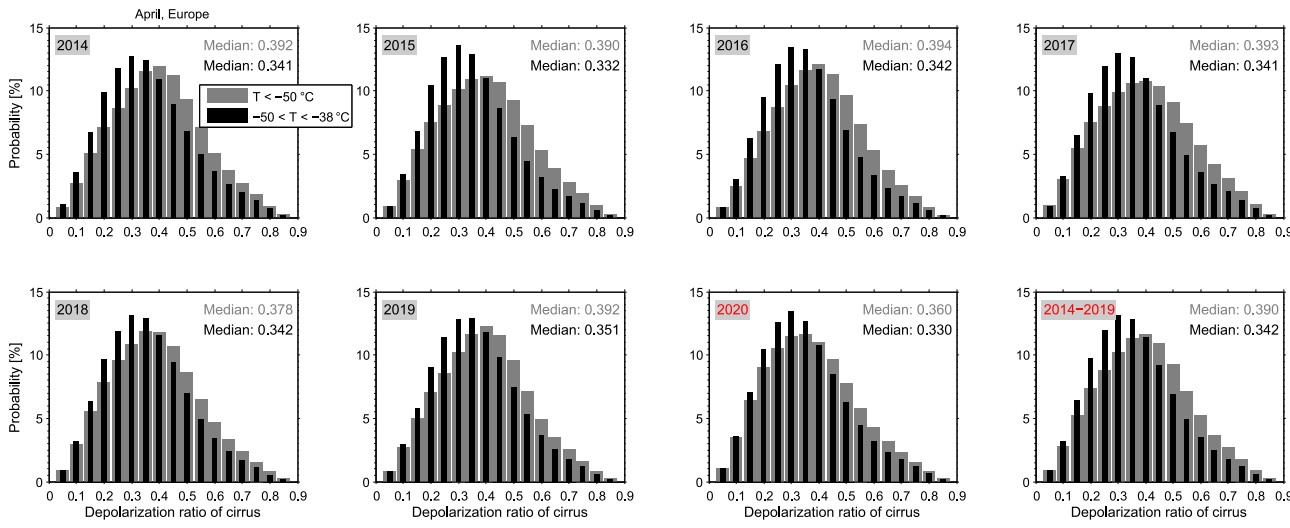

**Figure 8.** Distributions of the PLDR of cirrus clouds detected at temperatures from -50 °C to -38 °C (black) and at temperatures colder than -50 °C (grey). The corresponding medians for each case are indicated on the plot.

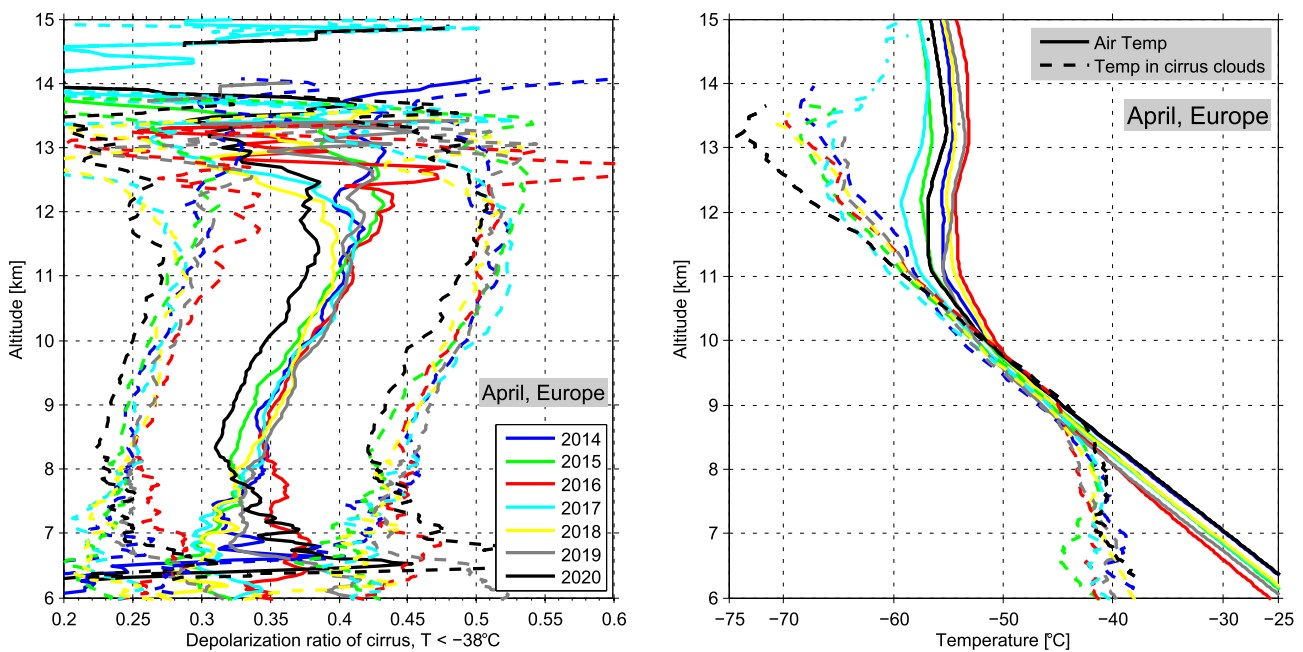

**Figure 9.** Altitude profiles of the medians of cirrus PLDR (solid lines) and their corresponding 25th and 75th percentiles derived from the CALIPSO observations in April at temperatures below -38 °C within the whole altitude range between 6 and 15 km (Left panel). For all the profiles, resulting PLDR medians show in general an increase with increasing altitudes from 8 to 12 km, i.e., the aviation cruising altitudes. The corresponding temperatures in cirrus clouds and the air temperatures of the background are shown with the dashed lines and solid lines, respectively, in the right panel.

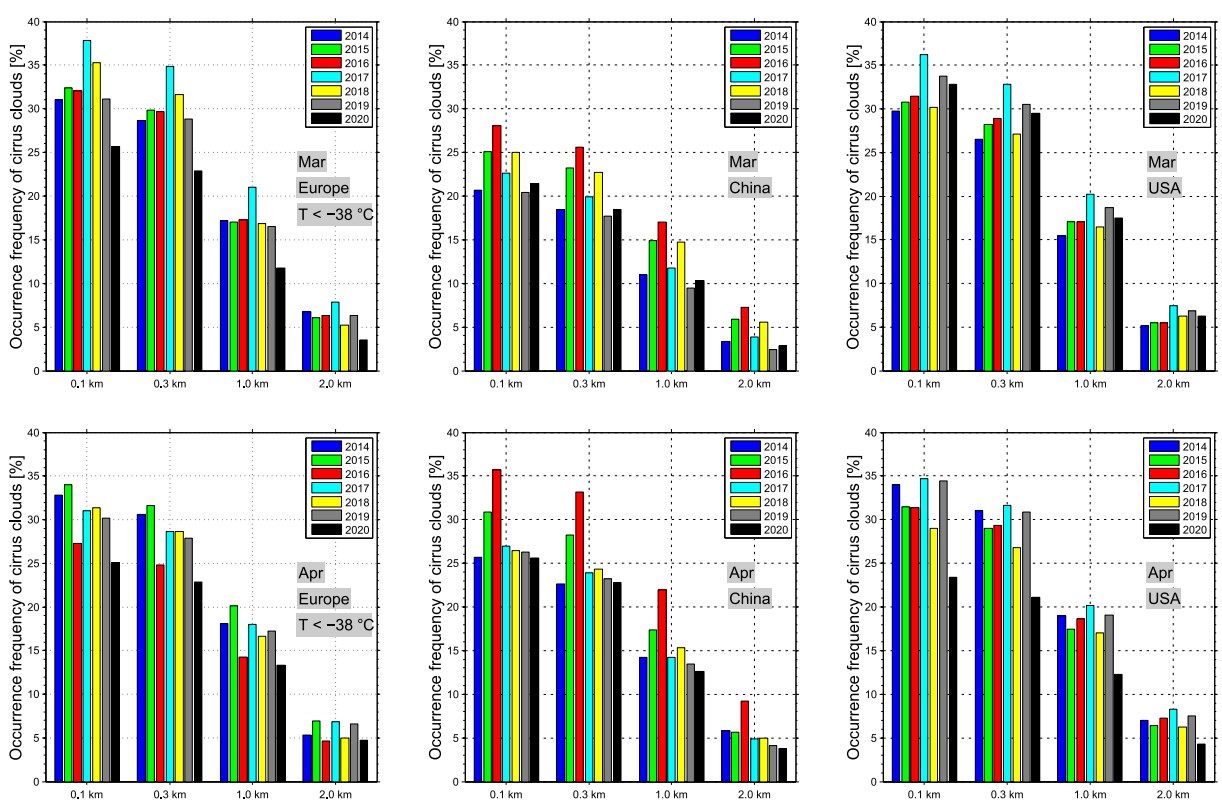

**Figure 10.** Histograms of the cirrus occurrence rates in years from 2014 to 2020 over Europe (left panels), the United States of America (middle panels), and China (right panels) with the definitions based on different cloud thicknesses larger than 0.1, 0.2, 1.0, and 2.0 km, respectively. Upper panels for the observations in March; Lower panels for April.

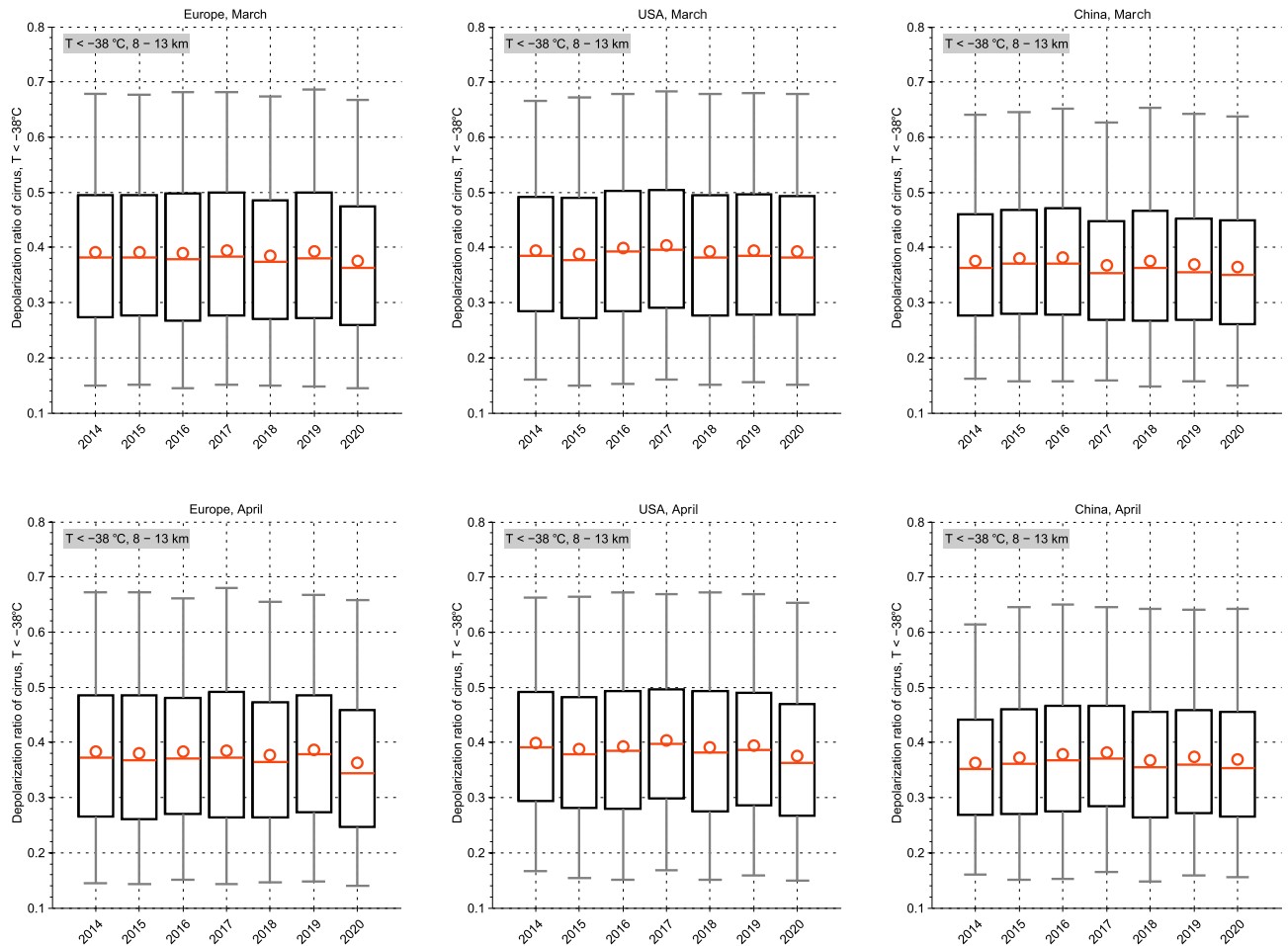

**Figure 11.** Boxplot representations of the PLDR of cirrus clouds detected in years from 2014 to 2020 over Europe (left panels), the United States of America (middle panels), and China (right panels). Upper panels for the observations in March; Lower panels for April. Boxes represent 25th–75th percentiles (top and bottom) and the solid lines in red through the corresponding boxes stand for the medians and red circles for means; whiskers in grey indicate the 5th and 95th percentiles and outliers which are larger than the upper whisker or smaller than the lower whisker are not shown here.