# Peer review of "Changes of cirrus cloud properties and occurrence over Europe during the COVID-19 caused air traffic reduction"

_Atmospheric Chemistry and Physics, 2021_

## Referee Comment (RC2)

Review of

**Changes of cirrus cloud properties and occurrence over Europe during the COVID-19 caused air traffic reduction**

Qiang Li and Silke Groß

**General:** Based on CALIPSO satellite lidar measurements, changes of cirrus cloud properties and occurrence caused by the air traffic reduction in March-April 2020 are investigated in this study. For this purpose, the meteorological situations of March-April of previous years were first analyzed in order to find years that are comparable to 2020, so that differences found between these years and 2020 can be clearly atrributed to the reduced air traffic. The findings presented are reduced cirrus cloud occurrences and thicknesses and also smaller mean values of cirrus linear depolarization ratio, especially at temperatures colder than T < -50◦ C.

The study fits well into the scope of ACP, is very interesting and of high relevance, especially from the perspective of ongoing anthropogenic impact on climate. The manuscript is clearly structured and fluently written, though unfortunately the phrasing is confusing at times, to my feeling due to language difficulties. Therefore, I recommend a language check.

Regarding the methods used in the study, I regret to say that I do not consider them to be scientifically mature. First, the physical understanding of cirrus and contrail formation seems to need some improvement (see comments in the introduction). Next, the analysis of the similarity of the years: it is performed for March to April, but in the following analysis March and April are treated seperately. I would recommend an analysis as in Figure 1 for both months and then a separate choice of years similar to 2020.

Here are some more examples of what I mean by a not mature analysis (more explanations are given in the specific comments): the years similar to 2020 are defined, but in the analysis one other year is used; Table 2 is not very informative; the estimate of the overall reduction in OR (cirrus occurrence rate) is too high, etc.

However, my major problem with the paper are the further analyzes and interpretations. First, the OR for all cirrus are considered. But when analyzing the PLDR (particle linear depolarization ratio), two temperature ranges are introduced, namely the one in which contrails can develop (at temperatures below about -50◦ C where the Schmidt- Appleman criterion is fulfilled) and the other at warmer temperatures, i.e. where the influence of aviation is at the most very small, but most likely not existing. I would say that if the reduction in cirrus is caused by air traffic, then this should only be seen in the cold temperatures, for both OR and PLDR. If the reduction is to be found at all altitudes (which seems to be the case from what I see in the presented material), this points to another reason. I can imagine that using years to compare with 2020 for March and April seperately would give clearer results. I also suggest to perform the analysis of both OR and PLDR for the two temperature ranges.

In summary, I would strongly encourage the authors to repeat the analysis by taking into account the recommendations outlined above and in the specific comments to present robust and important results on the aviation influence on cirrus occurrence and properties. After that, the article should be published in ACP.

The specific comments are organized as follows: Text from the manuscript ist shown in quotation marks, the comments are without.

**Specifc comments:**

**1) Page 1, lines 20-21:**
‚... ice crystals in air form and grow as a function of temperature and ice supersaturation ...'

The major driver is the vertical velocity.

**2) Page 1, lines 20-21:** ‚… there is a general trend toward larger morphological complexity as the supersaturation increases as well as the temperature drops...'

The morphological complexity decreases when the temperature drops. The colder it is, the more the ice particles tend towards spherical shapes, as there is not enough water available to form more complex shapes (see e.g. Lawson et al., 2019, JGR, Figure 22) or
https://www.researchgate.net/figure/Ice-crystal-shape-as-a-function-of-formation-temperature-and-supersaturation-with-respect_fig11_221917565

[Figure]

**3) Page 2, lines 5-6:** ‚According to theoretical ray-tracing simulations of laser backscatter depolarization (e.g. Takano and Liou, 1989), the geometric properties (shape and composition) of aerosols and ice crystals...'

Composition is not a geometric property.

**4) Page 2, lines 16-18:** '… to study the characteristics of ice clouds (e.g. Schotland et al., 1971; Sassen, 1991; Ansmann et al., 2003; Groß et al., 2012; Urbanek et al., 2018).

Please add here: Rolf et al. (2012) and Kienast-Sjögren et al. (2016).

Rolf, C., Krämer, M., Schiller, C., Hildebrandt, M., and Riese, M.: Lidar observation and model simulation of a volcanic-ash-induced cirrus cloud during the Eyjafjallajökull eruption, Atmos. Chem. Phys., 12, 10281–10294, https://doi.org/10.5194/acp-12-10281-2012, 2012.

Kienast-Sjögren, E., Rolf, C., Seifert, P., Krieger, U. K., Luo, B. P., Krämer, M., and Peter, T.: Climatological and radiative properties of midlatitude cirrus clouds derived by automatic evaluation of lidar measurements, Atmos. Chem. Phys., 16, 7605–7621, https://doi.org/10.5194/acp-16-7605-2016, 2016.

**5) Page 2, lines 31-32:** 'This region was also used in a recent study by Schumann et al. (2021) to investigate air traffic and contrail changes during COVID-19.'

Better (because regions cannot be used ...):
Recently, Schumann et al. (2021) investigated air traffic and contrail changes during COVID-19 in the same region.

**6) Page 2, lines 33-34:** 'To largely exclude the effect of meteorological conditions on cirrus occurrence and cirrus properties in our study, we extended this study to a larger ...'

At this point, it is not clear to the reader what your study is about .... this would have to be explained beforehand. I suggest to move the next paragraph ('During the COVID-19 pandemic aviation ...') before the sentence mentioned above ('This region was also used ...').
Also, briefly explain which methods are applied by Schumann et al. (2021) to clearly show the differences/similarities of the two studies.

**7) Page 3, lines 4-5:** ' In May/June 2020 aviation shows ...'

**8) Page 5, lines 11-12:** 'Cirrus ice crystals generally form  in  regions of ascending motions (producing the necessary supersaturation  ice) by ice nucleation on aerosol particles in the upper troposphere (in-situ origin cirrus), or they appear in the cold outflow of frontal systems or convection as frozen cloud droplets that had formed at lower altitudes and warmer temperatures (liquid origin cirrus).'

The sentence was confusing, see the changes.

**9) Page 5, lines 12-13:** 'Aircraft emission of aerosols can lead to the formation of contrails depending on the surrounding meteorological conditions (including temperature, pressure, and humidity)  at the flight track.'

The contrails form on the aerosols emitted by aircraft, but not these aerosols lead to the formation, but the water vapor from the aircraft together with the surrounding meteorological conditions.

**10) Page 5, lines 15-16:** 'In addition, the aerosols might also change the optical properties of naturally occurring cirrus clouds.'

Not the aerosols, but the appearing contrail or contrail-cirrus ice crystals might change the optical properties of naturally occurring cirrus clouds.

**11) Page 6, lines 1-2:** 'However, looking at the year to year variability for the time period May-August ...'
It would be good to show this, if not in the main article, then in an appendix or supplementary material.

**12) Page 6, lines 3-5:** 'For the relative humidity we found that the relative humidity for those ranges ...'
(a) Avoid repetition of 'relative humidity'
(b) What ranges do you mean here ?
(c) Is this sentence at the right place here? The text continues with the
   'The largest agreement of the general situation in March/April 2020 ...',
   Which general situation do you mean? The relative humidity is not shown … or do you refer to
   Figure 1?
(d) You continue with
   'Looking at the median and mean values of the general distribution ...'
   Which 'general distribution' do you mean here ? Relative humidity ?
Please clearly indicate what you describe here and, if it is relative humidity, it would be also good to show a figure comparable to Figure 1, if not in the main article, then in an appendix or supplementary material.

**13) Page 6, lines 10-11:** Only now you mention Figure 2, containing information on relative humidity, which would be needed already above (see last comment):
'The derived temperature and humidity along with their median and mean values in April are shown in Figure 2.'
But, why only April and not March-April is shown here ?
In lines 20-21 you mention again that 'So, we use CALIPSO data of March and April ...'
Or, better, show both months. And also, analyze both month seperately in Fig. 1 (see general comments).

Same question for Table 2 (note also that 'Medain' should be 'Median' in the temperature column). And, Table 2 is redundant to Figure 2.

It would be much more infomative (and in the following for the reader easier to follow the discussion) to list the deviation of the years with respect to 2020 and then highlight the choice of the years for comparison with 2020. This choice can also be noted in the Table caption.

| *2014* | *2015* | 2016 | 2017 | 2018 | *2019* | |
|---|---|---|---|---|---|---|
| *0.542* | *-0.264* | 1.827 | -1.100 | 0.895 | *0.197* | median T (C), difference to 2020 |

**14) Page 6, line 16:** '… a larger spread as found for the relative humidity.  The values differ ...'

**15) Page 6, line 20:** ' … we mainly focus our analysis to the years 2014, 2017, 2019 and 2020.'
I think you mean 2014, **2015** (not 2017), 2019 and 2020, yes (see line 17)?

2017 is quite different to 2020 in median T (see Table above), however, in Figures 3 (and following Figures) also 2017 is mentioned, though from Table 2 and line 17 2015 is closer to 2020 ? Is this a typo or did you analyze 2017 ? If this is the case, I recommend to redo the analyzes for the year 2015.

**16) Page 7, lines 7-8:** 'Even if we extend our examination (not shown) to the years 2015 … '

?? See previous comment – 2015 is quite close to 2020.

**17) Page 7, line 9:** 'And even for 2018 ….'

The largest difference of median T (and mean relative humidity) to 2020 is in 2016.

**18) Page 7, lines 16-17:** 'The cirrus OR for a geometrical thickness of 2.0 km shows only minor reduction of overall about 4% compared to more than 5% of the reference years.'

Calculating the percentage reduction of the geometrical thickness categories results in ~ 20 – 25 % for all  categories; I do not see a minor reduction for  geometrical thickness of 2.0 km.

> 0.1 km   19.4 %   (from 31% to 25%)
> 0.3 km   21.4 %   (from 28% to 22%)
> 1.0 km   25.0 %   (from 16% to 12%)
> 2.0 km   20.0 %   (from   5% to  4%)

**19) Page 7, lines 18-19:** 'From the current analysis, it is striking to note that the cirrus OR in April 2020 are smaller by a factor of 30% ...'

From the numbers in Table 3 it would be smaller than ~ 20 – 30% (see also previous comment). I recommend to also cahnge that in the abstract and other places in the manuscript.

**20) Page 7, lines 18-19:** 'The average thickness of cirrus clouds in April 2020, however, is significantly smaller and reduced to only 1.18 km.'

Do you have an explanation as to why that is ?

**21) Page 8, line 12:** 3.2   Cirrus Pparticle linear depolarization ratio

**22) Page 8, line 24:** 'T = -50∘ C is one of the threshold conditions for contrail formation ...'

If there is no contrail formation at warmer tempratures, then these clouds are most likely not influenced by aviation. I would think that then it would make sense to do the previous analyzes (Section 3.1) in addition also for  T < -50∘ C, as you do it now for the PLDR, yes ? Otherwise possible effects from the warmer natural cirrus are mixed in the those of contrail cirrus.

I also recommend to show the temperature (maybe median temperature at altitude intervals) on the right y-axis in Figure 3, left panel, to see the region of aviation influence.

**23) Page 9, line 2:** condistions =  conditions

**24) Page 9, line 2:** 'We also compare the vertical profiles of the PLDR median (Figure 7 - solid lines) along with the corresponding 20% and 80% percentiles (dashed lines) for the height range between 8 and 12.3 km. These are the typical cruising altitudes for passenger and cargo aircrafts. '

In Figure 7,  I also recommend to introduce the median temperature at the right y-axis. In addition, the lower altitides should also be shown to see the PDLR below the cruising altitude.

From Figure 6, it is seen that the median  PDLR at warmer temperatures is also lower  in 2020  than in the other years. Since this is below the main cruising altitudes, does that not indicate that the natural cirrus clouds will also be reduced in 2020, perhaps due to the meteorology? That should be discussed here.   (see also next comment)

**25) Page 9, line 24 ff:**    Test of the significance  of differences between the cirrus PLDR in different years:  I guess that  you apply  the test to  the whole  temperature range, yes ?  Given the differences of the   PLDR in the two ranges (<> -50◦ C) discussed in the previous section, I recommend to carry out  the tests for the two regions seperately, especially because the influence  of aviation   should be visible in particular at the colder temperatures as this is the   place where contrail / cotrail cirrus can form.
In case you would find a difference at warmer temperatures,  that would be an indication of a reason other than aviation.     (see also last comment)

Tables 4 and 5:   Extend the captions, for example explain which parameter the test was applied to and what  the meaning of p and h is.

**26) Page 11, line 6 ff: '...** the occurrence rates (OR) of cirrus clouds over these three regions show that on average the cirrus clouds occurred more frequently over Europe and USA than over China. ...'

The OR  of cirrus clouds over these three regions are discussed, but not shown. I recommend to present plots in an Appendix or Supplementary material.

**27) Page 11- 12:**   PLDR in different regions:  Are the meteorological conditions in the compared years also comparable over USA and China ?

**28) Page 12,  lines 19-20:** 'Due to the westerly jet stream, aerosol source is dominated by clean marine for the north Atlantic and European region and for the north American region, whereas by continental and dust for the Chinese region.'

I don't see a connection between this sentence and the previous conclusions ? … I would delete it.

**29) Figure 1:** Please indicate the unit in the color bar.

**30) Figure 5:**  Please include a color bar.
Caption:  '… more red area indicating larger number densities.'  Please check the language.

---

## Author Comment (AC1)

**Response to Review RC1 by Referee #1.**

We thank the Referee for his/her careful review and for the suggestions for an extended analysis. In the following, the Referee's questions and comments are repeated in black and our responses follow in blue.

**Summary of Paper**

The authors examine changes in cirrus cloud properties during the civil air traffic slowdown in March and April 2020 (due to COVID-19 air travel restrictions). CALIPSO vertical feature mask (VFM) data over western Europe are compared with data from similar periods from 2014 through 2019, and mean cirrus properties including cirrus cloud occurrence, average thickness, and particle linear depolarization ratio (PLDR) are determined for both 2020 and the earlier years. Cirrus cloud occurrence was found to decrease 30 percent in 2020 compared to three other years (2014, 2017, 2019) with no air traffic reductions and similar meteorological conditions. The calculated average cloud thickness was also less in 2020 (1.18 km) than in the previous years (1.40 km), and the PLDR values were reduced during air traffic slowdown. CALIPSO observations over China and USA were also examined to confirm the impact of air traffic reductions on cirrus cloud properties.

**General Comments**

Although the authors provide some solid evidence that cirrus cloud properties changed during the COVID-19 induced air traffic slowdown, several confusing aspects of the paper's presentation detract from the paper's value.

For example, the authors present bi-monthly (March and April) in Figures 1 and 8, but only monthly (April) data for the remaining figures. Are the authors conflating the March and April data as one distinct period of air traffic activity? As far as I can tell, air traffic volume was changing throughout March over Europe, so conditions between the two months may not be as similar as presented in the paper.

➔ This is right that air traffic started to reduce from about Mar.10, 2020 and reach the lowest level until Mar.23, 2020 (nearly 80% lower than 2019). While air traffic kept at that level during the whole month in April 2020 (Source: https://www.eurocontrol.int/Economics/2020-DailyTrafficVariation-States.html). Following the comments, we changed the presentation of 500 hPa geopotential height for a monthly one in March and April, respectively.

Second, it is not clear why or how the years 2014, 2017 and 2019 are chosen as the closest analogues to 2020, Those years might be the closest to 2020 in terms of mean 500-hPa geopotential height, but each is noticeably different. If the bi-monthly (or monthly, it's not clear which is used) mean meteorological properties are sufficient for judging the similiarity of meteorological conditions for each year, why not use the 6-year (2014 through 2019) mean instead? The 6-year mean appears to be the best match to 2020 in Figure 1.

➔ The reviewer is right, that we choose these years based on the 500-hPa GPH. We now included also different mean meteorological parameters in our consideration (e.g.

Temperature, RHi and vertical velocity). We see that the different years are quite different and show a large variability in the meteorological situation. In March, 2018 and 2020 are the outliers with higher temperatures and lower humidity at altitudes above 10 km. The difference of meteorology in 2018 can be also seen in the 500 hPa GPH. In April, however, the only 'outstanding' year is 2016 with warmer and drier airmass above 9 km and 2020 is well within the spread considering mean meteorological parameters.

➔ To avoid any misunderstanding, we furthermore extended out analysis for all the years from 2014 to 2020 for our comparison in the revised manuscript.

Third, which data source are the authors using to decide the proper analogues? The temperature and humidity data from Figure 2 suggest some meteorological differences between GEOS-5 and the NCEP/NCAR reanalyses from Figure 1 (which is not surprising), so which do the authors consider to be more reliable?

➔ The reviewer is right that temperature and humidity data from Figure 2 and the 500 hPa GPH in Figure 1 may show inconsistency since the meteorological data (temperature and humidity) were derived from different pressure levels. For the original manuscript, we compared the temp and humidity data from GEOS-5 because the data are saved in the CALIPSO data and hence easy to use. In the revised manuscript, we compare the meteorological data (temp, RHi, and vertical velocity) derived from ERA-5 to give a more general picture along the altitude range covering our observations. Anyway, we have extended our analysis to all the data from 2014 to 2020 for a more comprehensive comparison with a spread of meteorological variations.

Fourth, why include years 2015, 2016, and 2018 in Table 3 but not elsewhere in the paper? I believe the paper would be improved if the authors addressed these ambiguities, and perhaps only presenting data from April.

➔ Thank you for the suggestion. In the revised manuscript, we now include all the years from 2014 to 2020 in our analysis. In order to make the description in the manuscript smooth, we will present only the April results (cirrus occurrence and PLDR) and mention the March results in text and tables. Although the meteorological conditions are quite different, the results still strongly support our perspectives. Only the meteorological conditions in 2016 led to a stronger reduction in cirrus cloud occurrence. However, the cirrus clouds in 2016 did not show the reduction in PLDR as it was found in 2020.

Figures 3 and 4 are also unclear. The results of Figure 4 seem to contradict Figure 3! Figure 4 implies that the cirrus thickness distribution is skewed more toward thinner (less than 1 km) clouds in 2020 when compared to the other three years, yet Figure 3 shows that the decrease in occurrence rates in 2020 compared to other years is the largest for the thinner clouds. How is that possible? How can the relative frequency of thin clouds increase in 2020 while the occurrence of thin clouds decrease the most (compared to all other thicknesses) in 2020?

➔ We show the cirrus thickness distribution in relative frequency (100% in total). Here, the strong reduction of cirrus thickness with a mean of 1.18 km was found in March 2020 (instead of April, sorry for the confusing). We showed the cloud thickness in both March and April in the revised manuscript. The reduction in March is supposed to be stronger than expected from only the aviation change, since the meteorological conditions with

warmer and drier airmasses in March 2020 were less favorable for ice cloud formation than the previous years.

The discussion about the PLDR data over China and USA seems to be speculative, especially in terms of the meteorological conditions over both regions and in comparison to Europe. Do the authors know that USA and China have similar meteorology in 2014, 2017, and 2019 compared to 2020? If not, then Figure 8 implies that meteorology is not important for determining PLDR!

➔ The reviewer is right, we do not check if the same years can also be used for a comparison between 2020 and the selected years (2014, 2017, and 2019). We hence extended our analysis to the previous years from 2014 to 2019 without pre-selection. However, we agree, that the meteorological conditions are not important for the changes in cirrus PLDR.

➔ Furthermore, following the recent results from *Righi et al.* (2021), we would also not expect strong changes over the China region as the aviation effect on cirrus clouds in this region is in general very low.

Do the authors have any air traffic data to support the claims on lines 3-10 of page 12? Several factors are claimed to affect PLDR in the discussion (cirrus cloud height, the magnitude of air traffic, meteorological conditions) but these effects at best only described vaguely in this section.

➔ Yes, the sources of air traffic data we refer to are from www.airlines.org/dataset/ and www.eurocontrol.int/covid19 (last access: 25 June 2021).

P12,L3-5: "We next focus on the results observed over USA and see slightly larger values in 2017 for both months which may be due to the variations of meteorological conditions in different years and is comprehensible." Have the authors checked this claim to be sure? Like much of this section, the discussion here is vague and speculative.

➔ The reviewer is right! We reformed the discussion.

**Typographical errors and minor objections**

P4,L23: What are "radiative forces"?

➔ Sorry for the typo (radiative forcing). Changed.

P7, L2-3: The research area has already been defined earlier in the

manuscript so the mention of the lat/lon box is here is superfluous.

➔ Thank you. We reformed the sentence accordingly.

P7,L9: Please use higher altitude rather than larger altitude

➔ Thank you. Changed.

P9, L15-16: "But the decreases of the PLDR with height were only found

at altitudes larger ~10km in 2020." Is this referring to Mar 2020

only? This seems to contradict Figure 7.

➔ Yes, this is the results of March 2020 (not shown).

Several typographical errors were noticed in the text. Some of them are listed below although this is not exhaustive (please proofread the paper).

➔ Thank you. The typos have been corrected.

P3,L16 Constellatin

P4,L22 comsisting

P6,L8: Observering

P6,L20: propierties

P7,L9: differes

Table 2, column 2: Medain

P11,L14: quartiel

P12,L26: ocurrence

P13,L3: funciton

P13,L4: referece

P13,L5: charaterized

P13,L7: yeras

---

## Author Comment (AC2)

**Response to Review RC2 by Referee #2.**

We thank the Referee for carefully reading our manuscript and for the suggestions and specific comments to improve the current work. In the following the Referee's comments are repeated in black and our responses follow in blue.

Review of

**Changes of cirrus cloud properties and occurrence over Europe during the COVID-19 caused air traffic reduction**

Qiang Li and Silke Groß

**General**: Based on CALIPSO satellite lidar measurements, changes of cirrus cloud properties and occurrence caused by the air traffic reduction in March-April 2020 are investigated in this study. For this purpose, the meteorological situations of March-April of previous years were first analyzed in order to find years that are comparable to 2020, so that differences found between these years and 2020 can be clearly attributed to the reduced air traffic. The findings presented are reduced cirrus cloud occurrences and thicknesses and also smaller mean values of cirrus linear depolarization ratio, especially at temperatures colder than T < -50◦ C.

The study fits well into the scope of ACP, is very interesting and of high relevance, especially from the perspective of ongoing anthropogenic impact on climate. The manuscript is clearly structured and fluently written, though unfortunately **the phrasing is confusing at times**, to my feeling due to language difficulties. Therefore, I recommend a language check.

➔ Thank you.

Regarding the methods used in the study, I regret to say that I do not consider them to be scientifically mature. First, the physical understanding of cirrus and contrail formation seems to need some improvement (see comments in the introduction). Next, the analysis of the similarity of the years: it is performed for March to April, but in the following analysis March and April are treated separately. I would recommend an analysis as in Figure 1 for both months and then a separate choice of years similar to 2020.

➔ We agree that it would be better to treat the two months separately in the meteorological discussion. We accordingly changed the meteorological analysis. We furthermore added more parameters in this discussion. We find a large variability in the meteorological conditions for the years from 2014 to 2019 with the conditions of 2020 in both March and April well within the spread. In March, 2018 and 2020 are the outliers with higher temperatures and lower RHi at altitudes above 10 km. In April, however, the only 'outstanding' year is 2016, which showed significantly higher temperature and lower RHi values. We furthermore extended our analysis for all the years from 2014 to 2019 for our comparison. The results provide strong support to our statement that the cirrus cloud cover and properties did not change in the former years but show significant reduction in March/April 2020. Only the meteorological conditions in 2016 led to a stronger reduction in cirrus cloud occurrence. However, the cirrus clouds in 2016 did not show the reduction in PLDR as it was found in 2020.

Here are some more examples of what I mean by a not mature analysis (more explanations are given in the specific comments): the years similar to 2020 are defined, but in the analysis one other year is used; Table 2 is not very informative; the estimate of the overall reduction in OR (cirrus occurrence rate) is too high, etc.

However, my major problem with the paper are the further analyzes and interpretations. First, the OR for all cirrus are considered. But when analyzing the PLDR (particle linear depolarization ratio), two temperature ranges are introduced, namely the one in which contrails can develop (at temperatures below about -50∘ C where the Schmidt- Appleman criterion is fulfilled) and the other at warmer temperatures, i.e. where the influence of aviation is at the most very small, but most likely not existing. I would say that if the reduction in cirrus is caused by air traffic, then this should only be seen in the cold temperatures, for both OR and PLDR. If the reduction is to be found at all altitudes (which seems to be the case from what I see in the presented material), this points to another reason. I can imagine that using years to compare with 2020 for March and April separately would give clearer results. I also suggest to perform the analysis of both OR and PLDR for the two temperature ranges.

➔ We agree with this Reviewer that not only PLDR should be considered for the two temperature ranges but also for the OR. Thank you for the suggestions. We changed our analysis accordingly and found that a clear reduction in cirrus OR was found at temperatures below -50 °C, while at warmer temperatures cirrus OR in April 2020 were slightly smaller than in the reference years at lower altitudes below 9 km and became close to and even slightly larger than the other years at higher altitudes.

In summary, I would strongly encourage the authors to repeat the analysis by taking into account the recommendations outlined above and in the specific comments to present robust and important results on the aviation influence on cirrus occurrence and properties. After that, the article should be published in ACP.

➔ We thank this Reviewer for his/her valuable comments and repeated our analysis accordingly. Please find the specific answers following the different suggestions.

The specific comments are organized as follows: Text from the manuscript is shown in quotation marks, the comments are without.

**Specifc comments**:

1) Page 1, lines 20-21: ‚... ice crystals in air form and grow as a function of temperature and ice supersaturation ...' The major driver is the vertical velocity.

➔ We agree, that the vertical velocity was missing. We added this and also analyzed the vertical velocity in our meteorological analysis. Although there was a large variability in the profiles of vertical velocity in different years (2014, 2015, 2017, and 2019), cirrus occurrence did not show clear difference. The specific case (such as April 2016) showed that cirrus occurrence is more dependent on the temp and RHi than vertical velocity.

2) Page 1, lines 20-21: ‚… there is a general trend toward larger morphological complexity as the supersaturation increases as well as the temperature drops...'

The morphological complexity decreases when the temperature drops. The colder it is, the more the ice particles tend towards spherical shapes, as there is not enough water available to form more complex shapes (see e.g. Lawson et al., 2019, JGR, Figure 22) or https://www.researchgate.net/figure/Ice-crystal-shape-as-a-function-of-formation-temperature-andsupersaturation-with-respect_fig11_221917565

➔ We followed this suggestion and reworked this paragraph in the manuscript. But the cirrus depolarization ratios increase as the temperatures drops based on in-situ measurements (e.g., Urbanek et al., 2018).

3) Page 2, lines 5-6: ,According to theoretical ray-tracing simulations of laser backscatter depolarization (e.g. Takano and Liou, 1989), the geometric properties (shape and composition) of aerosols and ice crystals...' Composition is not a geometric property.

➔ Right! This should be 'shape and size'. We changed that accordingly.

4) Page 2, lines 16-18: '... to study the characteristics of ice clouds (e.g. Schotland et al., 1971; Sassen, 1991; Ansmann et al., 2003; Groß et al., 2012; Urbanek et al., 2018). Please add here: Rolf et al. (2012) and Kienast-Sjögren et al. (2016). Rolf, C., Krämer, M., Schiller, C., Hildebrandt, M., and Riese, M.: Lidar observation and model simulation of a volcanic-ash-induced cirrus cloud during the Eyjafjallajökull eruption, Atmos. Chem. Phys., 12, 10281–10294, https://doi.org/10.5194/acp-12-10281-2012, 2012. Kienast-Sjögren, E., Rolf, C., Seifert, P., Krieger, U. K., Luo, B. P., Krämer, M., and Peter, T.: Climatological and radiative properties of midlatitude cirrus clouds derived by automatic evaluation of lidar measurements, Atmos. Chem. Phys., 16, 7605–7621, https://doi.org/10.5194/acp-16-7605-2016, 2016.

➔ Thank you for this comment, we added the suggested references.

5) Page 2, lines 31-32: 'This region was also used in a recent study by Schumann et al. (2021) to investigate air traffic and contrail changes during COVID-19.' Better (because regions cannot be used ...): Recently, Schumann et al. (2021) investigated air traffic and contrail changes during COVID-19 in the same region.

➔ Thank you for the suggestions. We reformed the sentence accordingly.

6) Page 2, lines 33-34: 'To largely exclude the effect of meteorological conditions on cirrus occurrence and cirrus properties in our study, we extended this study to a larger ...' At this point, it is not clear to the reader what your study is about …. this would have to be explained beforehand. I suggest to move the next paragraph ('During the COVID-19 pandemic aviation ...') before the sentence mentioned above ('This region was also used ...'). Also, briefly explain which methods are applied by Schumann et al. (2021) to clearly show the differences/similarities of the two studies.

➔ Thank you for the suggestions. We reformed the paragraph accordingly and updated the results from literature.

7) Page 3, lines 4-5: 'Since In May/June 2020 aviation shows ...'

➔ Revised, thank you.

8) Page 5, lines 11-12: 'Cirrus ice crystals generally form in the outflow frontal of the deep convections and in the regions of ascending motions (producing the necessary supersaturation

of over ice), or they form by ice nucleation on aerosol particles in the upper troposphere (in-situ origin cirrus), or they appear in the cold outflow of frontal systems or convection as frozen cloud droplets that had formed at lower altitudes and warmer temperatures (liquid origin cirrus).' The sentence was confusing, see the changes.

➔ Changed, thank you.

9) Page 5, lines 12-13: 'Aircraft emission of aerosols can lead to the formation of contrails depending on the surrounding meteorological conditions (including temperature, pressure, and humidity) of at the flight track.' The contrails form on the aerosols emitted by aircraft, but not these aerosols lead to the formation, but the water vapor from the aircraft together with the surrounding meteorological conditions.

➔ Reformed.

10) Page 5, lines 15-16: 'In addition, the aerosols might also change the optical properties of naturally occurring cirrus clouds.' Not the aerosols, but the appearing contrail or contrail-cirrus ice crystals might change the optical properties of naturally occurring cirrus clouds.

➔ Changed.

11) Page 6, lines 1-2: 'However, looking at the year to year variability for the time period MayAugust ...' It would be good to show this, if not in the main article, then in an appendix or supplementary material.

➔ Thank you for the comment. The cirrus occurrence and optical properties show a clear seasonal variation. We would not show supplementary material here as we are working on a long-term comparison and won't publish part of the unfinished study in here.

12) Page 6, lines 3-5: 'For the relative humidity we found that the relative humidity for those ranges ...' (a) Avoid repetition of 'relative humidity' (b) What ranges do you mean here ? (c) Is this sentence at the right place here? The text continues with the 'The largest agreement of the general situation in March/April 2020 ...', Which general situation do you mean? The relative humidity is not shown … or do you refer to Figure 1? (d) You continue with 'Looking at the median and mean values of the general distribution ...' Which 'general distribution' do you mean here ? Relative humidity ? Please clearly indicate what you describe here and, if it is relative humidity, it would be also good to show a figure comparable to Figure 1, if not in the main article, then in an appendix or supplementary material.

➔ (a) thank you for pointing out; (b) altitude range of 8-13 km; (c) No, we refer to Figure 2 (right panel); (d) still relative humidity.
➔ Thank you for the comments and sorry for the misleading. In the revised manuscript, we will compare the altitude profiles of temp, RHi, and vertical velocity derived from ERA-5. We reformed this paragraph accordingly.

13) Page 6, lines 10-11: Only now you mention Figure 2, containing information on relative humidity, which would be needed already above (see last comment): 'The derived temperature and humidity along with their median and mean values in April are shown in Figure 2.' But, why only April and not March-April is shown here ? In lines 20-21 you mention again that 'So, we use CALIPSO data of March and April ...' Or, better, show both months. And also, analyze both month seperately in Fig. 1 (see general comments). Same question for Table 2 (note also that 'Medain'

should be 'Median' in the temperature column). And, Table 2 is redundant to Figure 2. It would be much more infomative (and in the following for the reader easier to follow the discussion) to list the deviation of the years with respect to 2020 and then highlight the choice of the years for comparison with 2020. This choice can also be noted in the Table caption. 2014 2015 2016 2017 2018 2019 0.542 -0.264 1.827 -1.100 0.895 0.197 median T (C), difference to 2020

➔ See comment above. In the revised manuscript, we changed the discussion on the meteorological conditions and included all the years from 2014 to 2019 in our comparison.

14) Page 6, line 16: '… a larger spread as found for the relative humidity. However, t The values differ …'

➔ See comment above. Figure 2 in the original manuscript was replaced by the results of ERA5.

15) Page 6, line 20: ' … we mainly focus our analysis to the years 2014, 2017, 2019 and 2020.' I think you mean 2014, 2015 (not 2017), 2019 and 2020, yes (see line 17)? 2017 is quite different to 2020 in median T (see Table above), however, in Figures 3 (and following Figures) also 2017 is mentioned, though from Table 2 and line 17 2015 is closer to 2020 ? Is this a typo or did you analyze 2017 ? If this is the case, I recommend to redo the analyzes for the year 2015.

➔ In the original manuscript, we selected the three years 2014, 2017, and 2019 as reference year according to the similar meteorological conditions. So there is no typo in using 2017. Thank you for the recommendation. We extended our analysis to all the years from 2014 to 2020 without pre-selection.

16) Page 7, lines 7-8: 'Even if we extend our examination (not shown) to the years 2015 … ' ?? See previous comment – 2015 is quite close to 2020.

➔ Please see the previous comment.

17) Page 7, line 9: 'And even for 2018 ….' The largest difference of median T (and mean relative humidity) to 2020 is in 2016.

➔ We reworked our discussion on that.

18) Page 7, lines 16-17: 'The cirrus OR for a geometrical thickness of 2.0 km shows only minor reduction of overall about 4% compared to more than 5% of the reference years.' Calculating the percentage reduction of the geometrical thickness categories results in ~ 20 – 25 % for all categories; I do not see a minor reduction for geometrical thickness of 2.0 km. > 0.1 km 19.4 % (from 31% to 25%) > 0.3 km 21.4 % (from 28% to 22%) > 1.0 km 25.0 % (from 16% to 12%) > 2.0 km 20.0 % (from 5% to 4%)

➔ Thank you for you comment, we changed that.

19) Page 7, lines 18-19: 'From the current analysis, it is striking to note that the cirrus OR in April 2020 are smaller by a factor of 30% ...' From the numbers in Table 3 it would be smaller than ~ 20 – 30% (see also previous comment). I recommend to also change that in the abstract and other places in the manuscript.

➔ Changed. Thank you!

20) Page 7, lines 18-19: 'The average thickness of cirrus clouds in April 2020, however, is significantly smaller and reduced to only 1.18 km.' Do you have an explanation as to why that is ?

➔ Here, the strong reduction of cirrus thickness with a mean of 1.18 km was found in March 2020 (sorry for the confusing). The reduction was supposed to be stronger than expected from the aviation change, since the meteorological conditions with warmer and drier airmasses in March 2020 were less favorable for ice cloud formation than the previous years.

21) Page 8, line 12: 3.2 Cirrus particle linear depolarization ratio

➔ Changed.

22) Page 8, line 24: 'T = -50∘ C is one of the threshold conditions for contrail formation ...' If there is no contrail formation at warmer tempratures, then these clouds are most likely not influenced by aviation. I would think that then it would make sense to do the previous analyzes (Section 3.1) in addition also for T < -50∘ C, as you do it now for the PLDR, yes ? Otherwise possible effects from the warmer natural cirrus are mixed in the those of contrail cirrus. I also recommend to show the temperature (maybe median temperature at altitude intervals) on the right y-axis in Figure 3, left panel, to see the region of aviation influence.

➔ It should be noted that the temperatures do not decrease monotonically with altitudes between 6 and 14 km covering the tropopause. It makes no sense to show the dependence of cirrus occurrence on temperatures, although we have done the analysis. Alternatively, we showed the cirrus occurrence at temperatures between -50 °C and -38 °C and below -50 °C, respectively, in the revised manuscript.

23) Page 9, line 2: condistions = conditions

➔ Changed. Thank you.

24) Page 9, line 2: 'We also compare the vertical profiles of the PLDR median (Figure 7 - solid lines) along with the corresponding 20% and 80% percentiles (dashed lines) for the height range between 8 and 12.3 km. These are the typical cruising altitudes for passenger and cargo aircrafts. ' In Figure 7, I also recommend to introduce the median temperature at the right y-axis. In addition, the lower altitides should also be shown to see the PDLR below the cruising altitude. From Figure 6, it is seen that the median PDLR at warmer temperatures is also lower in 2020 than in the other years. Since this is below the main cruising altitudes, does that not indicate that the natural cirrus clouds will also be reduced in 2020, perhaps due to the meteorology? That should be discussed here. (see also next comment)

➔ See the answer to the question 22). It is true that the PLDR values in 2020 also show a reduction, to a smaller extent, at warmer temperatures, which, however, were detected at the altitudes between 6 and 11 km. Namely, cirrus clouds at warmer temperatures occurred also partly within the aviation cruising altitudes. You can see this feature in the middle panel of Figure 6 in the revised manuscript. Besides inducing the formation of contrails, aircraft emission aerosols also cause indirect effects on naturally occurring cirrus by increasing heterogeneous nucleation.

25) Page 9, line 24 ff: Test of the significance of differences between the cirrus PLDR in different years: I guess that you apply the test to the whole temperature range, yes ? Given the

differences of the PLDR in the two ranges (<> -50° C) discussed in the previous section, I recommend to carry out the tests for the two regions seperately, especially because the influence of aviation should be visible in particular at the colder temperatures as this is the place where contrail / cotrail cirrus can form. In case you would find a difference at warmer temperatures, that would be an indication of a reason other than aviation. (see also last comment) Tables 4 and 5: Extend the captions, for example explain which parameter the test was applied to and what the meaning of p and h is.

➔ Yes, we did the significance test for the whole temperature range (T<-38°C), but for the observations at altitudes between 8 and 13 km (the aviation cruising altitudes). It's stated in the last answer that the cirrus clouds occurring at warmer temperatures also partly appear at the aviation cruising altitudes. We added the explanation of *p* and *h*.

26) Page 11, line 6 ff: '... the occurrence rates (OR) of cirrus clouds over these three regions show that on average the cirrus clouds occurred more frequently over Europe and USA than over China. ...' The OR of cirrus clouds over these three regions are discussed, but not shown. I recommend to present plots in an Appendix or Supplementary material.

➔ We added a figure to show the regional difference in the cirrus occurrence rates over the three areas.

27) Page 11- 12: PLDR in different regions: Are the meteorological conditions in the compared years also comparable over USA and China ?

➔ We did not analyze the meteorological conditions so far. So, we reworked this section. According to a lately published article by *Righi et al.* (2021) we would not expect changes in the China region, as aviation effects on cirrus clouds is low in this region.

28) Page 12, lines 19-20: 'Due to the westerly jet stream, aerosol source is dominated by clean marine for the north Atlantic and European region and for the north American region, whereas by continental and dust for the Chinese region.' I don't see a connection between this sentence and the previous conclusions ? … I would delete it.

➔ We reworked this section and deleted this sentence.

29) Figure 1: Please indicate the unit in the color bar.

➔ The unit in the color bar was added.

30) Figure 5: Please include a color bar. Caption: '… more red area indicating larger number densities.' Please check the language

➔ A color bar was added. We reformed the sentence.

---

## Author Response (AR2)

**Report #1**

Submitted on 30 Jul 2021
Anonymous Referee #2

**Anonymous during peer-review: Yes** No
**Anonymous in acknowledgements of published article: Yes** No

**Recommendation to the editor**

| | |
|---|---|
| **1) Scientific significance** Does the manuscript represent a substantial contribution to scientific progress within the scope of this journal (substantial new concepts, ideas, methods, or data)? | Outstanding Excellent **Good** Fair Low |
| **2) Scientific quality** Are the scientific approach and applied methods valid? Are the results discussed in an appropriate and balanced way (consideration of related work, including appropriate references)? | Outstanding Excellent **Good** Fair Low |
| **3) Presentation quality** Are the scientific results and conclusions presented in a clear, concise, and well structured way (number and quality of figures/tables, appropriate use of English language)? | Outstanding Excellent **Good** Fair Low |

For final publication, the manuscript should be
**accepted as is**
accepted subject to **technical corrections**
**accepted subject to minor revisions**
reconsidered after **major revisions**
**rejected**

**Were a revised manuscript to be sent for another round of reviews:**
**I would be willing to review the revised manuscript.**
I would not be willing to review the revised manuscript.

**Suggestions for revision or reasons for rejection (will be published if the paper is accepted for final publication)**

The authors have revised the manuscript significantly by redoing all the analysis following the suggestions given in the reviews. Thus, to my opinion the paper has greatly improved and is now physically sound and ready for publication.

> I have only one further remark. In the meantime another publication appeared dealing with the same topic:
>
> Climate impact of aircraft-induced cirrus assessed from satellite observations before and during COVID-19, Quaas et al. 2021, ERL.
>
> stating that
>
> ' Here we show, using an analysis of satellite observations for the period March–May 2020, that in the 20% of the Northern Hemisphere mid-latitudes with the largest air traffic reduction, cirrus fraction was reduced by ~9 ± 1.5% ...'
>
> How does that compare to your finding that: '... the cirrus cloud occurrence was reduced by about 17–30% ...'
>
> I recommend to include a comparison between the two studies in your final revised version.

We thank the Referee for the second-round review. To compare our study to the recent published results by Quaas et al. 2021, we note the difference as follows:

1. The difference of detect limit between MODIS and CALIOP: Ackermann et al. (2008) found that 90% of the failed detections occur when the cloud optical depth was less than 0.4; while CALIOP can comprehensively observe the thin cirrus clouds with optical depth from 0.01 to 5 (Winker et al. 2009; Fu et al. Sci. Rep., 2017). So, MODIS is not so good in detecting the very small clouds that actually have the largest changes. In the manuscript, we looked on the reduction of cirrus cloud occurrence depending on the vertical extend and larger changes were found for thinner cirrus (geometrical thickness < 1 km) whereas nearly no changes found for cirrus with large (>2km) vertical extend.

2. Different research area: we concentrated on the European regions covering part of the northern Atlantic flight corridor while they looked at a larger area. A global distribution of cirrus occurrence rate shows that there is a larger cirrus occurrence rate over Europe than the mean values of the northern hemisphere midlatitude (e.g., Sassen et al. JGR 2008).

3. It's mentioned in Quaas et al. 2021 that there is a linear trend (positive in general) in cirrus (fraction and emissivity) over the period 2011-2019. While our study focusing on the reference years 2014-2019 may lead to a larger reduction in 2020 compared with the reference years.

4. In the manuscript, we only looked into March and April data and as we see from the comparison of the two months with quite a large effect. We also compared the results of CALIPSO data in May, showing very little changes, if it's not no changes, in the cirrus occurrence rate and the geometrical thickness.

5. They considered also the dependence on the changes in air traffic (flight track density change), while our study focusing on a region with one of the largest changes (reduced air traffic by >80% in April).

Due to the differences stated above, this might not be comparable one by one. But a short comparison will be stated in the manuscript.

**Report #2**

Submitted on 09 Aug 2021
Anonymous Referee #1

**Anonymous during peer-review: Yes** No
**Anonymous in acknowledgements of published article: Yes** No

**Recommendation to the editor**

| | |
|---|---|
| **1) Scientific significance**
Does the manuscript represent a substantial contribution to scientific progress within the scope of this journal (substantial new concepts, ideas, methods, or data)? | Outstanding Excellent **Good** Fair Low |
| **2) Scientific quality**
Are the scientific approach and applied methods valid? Are the results discussed in an appropriate and balanced way (consideration of related work, including appropriate references)? | Outstanding Excellent **Good** Fair Low |
| **3) Presentation quality**
Are the scientific results and conclusions presented in a clear, concise, and well structured way (number and quality of figures/tables, appropriate use of English language)? | Outstanding **Excellent** Good Fair Low |

For final publication, the manuscript should be
**accepted as is**
accepted subject to **technical corrections**
**accepted subject to minor revisions**
reconsidered after **major revisions**
**rejected**

**Were a revised manuscript to be sent for another round of reviews:**
I would be willing to review the revised manuscript.
**I would not be willing to review the revised manuscript.**

**Suggestions for revision or reasons for rejection (will be published if the paper is accepted for final publication)**

The authors have improved the paper considerably. They have addressed my comments in the previous manuscript, strengthened their arguments and clarified their presentation in many ways. My only objection would be that the authors include very little discussion of any differences in the meteorological conditions over the US and China during 2020 compared to

> previous years. Otherwise, I consider the journal article to be a good paper. They have provided good evidence supporting the view that cirrus cloud properties over western Europe changed during the COVID-19 induced air traffic slowdown, and that those changes are caused by the reduction in aviation over Europe during the slowdown.

We thank the Referee for the second-round review and the positive comments. In the manuscript, we focused on the observations over European regions and found that the occurrence rate and thickness of cirrus clouds strongly depended on the meteorological conditions, while this is not the case for the PLDR of cirrus (see the results of April 2016 compared with other years). For the regional comparison, we mainly looked into the cirrus PLDR. The comparison of cirrus geometrical thickness (in occurrence frequency) is to indicate the difference of cirrus properties as well as the different effects of air traffic over China from the other two regions. We may extend the analysis on the regional comparison including meteorological condition comparison in a separate study.